



# Attenuation of Sound in Glacier Ice from 2 kHz to 35 kHz

Alexander Meyer[1], Dmitry Eliseev[1], Dirk Heinen[1], Peter Linder[1], Franziska Scholz[1], Lars Steffen Weinstock[1], Christopher Wiebusch[1], and Simon Zierke[1]

[1]III. Physikalisches Institut B, RWTH Aachen University, Otto Blumenthal Str., 52074 Aachen, Germany

**Correspondence:** Christopher Wiebusch (wiebusch@physik.rwth-aachen.de)

**Abstract.** The acoustic damping of sound waves in natural glaciers is a largely unexplored physical property that has relevance for various applications. We present measurements of the attenuation of sound in ice with a dedicated measurement setup *in situ* on the Italian glacier *Langenferner*. The tested frequency ranges from 2 kHz to 35 kHz and probed distances between 5 meter and 90 meter. The attenuation length has been determined by two different methods and detailed investigations of systematic uncertainties. The attenuation length decreases slowly with increasing frequencies. Observed values range between 13 meter for low frequencies and 5 meter for high frequencies.The here presented results strongly improve in accuracy with respect to previous measurements. However, quantitatively the found attenuation is remarkably similar to observations at very different locations.

## 1 Introduction

The acoustical properties of ice are of interest for a large variety of applications ranging from the measurement of seismic waves (Robinson, 1968) to the detection of ultra-high-energy neutrinos (Abbasi et al., 2010). Recently, the application of sonographic methods has received increased interest in the context of the exploration of subglacial lakes in Antarctica or even water oceans below the ice surfaces of moons in the outer solar system. Particularly the joint research collaboration *Enceladus Explorer*, (Kowalski et al., 2016) has developed a maneuverable melting probe in glacial ice. It incorporates two acoustic systems operating in the range of 1 kHz to 1000 kHz. One is based on trilateration of the arrival times of acoustic signals from pingers and allows for the localization of the probe. The other system is based on phased piezo arrays and is used for the sonographic fore-field reconnaissance e.g. the detection of obstacles on the planned trajectory or water pockets when approaching the region of interest.

In water, sonographic imaging and acoustic localization techniques are well established technologies. In ice, however, acoustic navigation techniques are largely unexplored though they may provide a number of applications. Unlike water, not only pressure waves but also shear waves can propagate in the solid state ice. Since pressure waves are easier to generate and have a faster propagation speed (Vogt et al., 2008; Abbasi et al., 2010), they seem more suited for navigation purposes and are focused on in the following.

A limiting parameter is the damping of acoustic signals with distance, that strongly depends on the respective glacial environment and the frequency of the signal. In the following we refer to the attenuation length as that distance $r$ at which the amplitude of a spherical signal is reduced by $1/e$ after correcting the amplitude for the $1/r$ reduction. This parameter itself is





an interesting physical property as it depends on both the structures on scales of the wave-length and smaller but effectively integrated over the overall glacial structure. For the purpose of navigation it ultimately limits the maximum distance to which pairs of receiver and emitters can exchange signals. The design and optimization of acoustic transducers of high emission power strongly depends on the frequency and prefers higher frequencies as well as a better beam resolution of phased arrays

does.

The acoustic attenuation length in ice is not well known in the range from $1\,\mathrm{kHz}$ to $100\,\mathrm{kHz}$. While in water the attenuation length in this frequency range exceeds orders of kilometers (Fisher and Simmons, 1977; Schulkin and Marsh, 1962) and only slightly varies with temperature and chemical composition, the attenuation in the solid state material ice is more complicated. Even for simple polycrystaline ice, calculations range over orders of magnitude from a few 10 meters to several kilometers

depending on the temperature and assumed grain sizes (Price, 2006, 1993).

In a natural glacier environment the situation is even more complicated. Ice cracks filled with air and inclusions of dust and rocks will attenuate sound strongly. Their occurrence depends the general environmental conditions of the glacier such as its formation and flow.

Only few *in situ* measurements exist in the literature for very different glacial environments. The largest measured attenuation

length is consistent with about $300\,\mathrm{m}\pm20\,\%$. It has been observed for the glacial ice at depths $190\,\mathrm{m}$ to $500\,\mathrm{m}$ below the surface at the geographical South Pole, for frequencies between $10\,\mathrm{kHz}$ to $30\,\mathrm{kHz}$. This attenuation is however substantially stronger than the earlier predictions (Price, 2006). Measurements in sea ice by Langleben (1969) for $10\,\mathrm{kHz}$ to $500\,\mathrm{kHz}$ resulted in the range of $9\,\mathrm{m}$ to $2\,\mathrm{m}$ for $10\,\mathrm{kHz}$ to $30\,\mathrm{kHz}$. For frequencies $>100\,\mathrm{kHz}$ see also Lebedev and Sukhorukov (2001). Mesurements of seismic explosion shocks in a temperate glacier are reported in Westphal (1965). These measurememts result

in an amplitude attenuation length that ranges between $70\,\mathrm{m}$ to $4.6\,\mathrm{m}$ for frequencies from $2.5\,\mathrm{kHz}$ to $15\,\mathrm{kHz}$. This strong frequency dependency is interpreted as Rayleigh scattering on ice grains as dominant attenuation process. Recent measurements on the alpine glaciers *Morteratsch* and *Pers* (Helbing et al., 2016; Kowalski et al., 2016) with acoustic transducers reported an attenuation of similar scale with a length of $31\,\mathrm{m}$ for $5\,\mathrm{kHz}$ and $15\,\mathrm{m}$ for $18\,\mathrm{kHz}$. Goal of this work has been to provide a robust measurement that properly addresses and reduces experimental uncertainties with respect to previous measurements.

The measurement of the sound attenuation of sound *in situ* is in fact challenging and the accuracy is limited by the quality of the measurement setup and the systematic uncertainties related to the environment. In particular two aspects are important. First, sensors and emitters are inserted into the glacier by holes. The structure of such holes depends on the production process. It differs from hole to hole and changes with time, e.g. because the water-level can change with time due to leakage and refreezing of the walls. As result, the acoustic coupling to the ice differs not only from hole to hole but also for repeated

measurements in the same holes. Secondly, the natural glacial environment contains cracks and other absorbing structures. The subsurface ice-structure is unknown. The phase of reflected signals e.g. from the surface, depends on the specific emitter-receiver measurement geometry and thus can interfere with the direct acoustic signal.

The basic concept of the here presented measurement addresses these issues. It is based on the deployment of an acoustic emitter and a receiver a few meter deep into the glacier using holes that are produced with a melting probe. From the relative

amplitude of the signal registered for different distances we can infer the attenuation length.



In order to produce an as robust result as possible, we have established the following strategy:

1. In all measurement the same pair of sender and receiver is used. Therefore the emitter and receiver sensitivities cancel in the ratio of received signals of different distances.

2. We use an emitter and a receiver that are largely spherically symmetric in emissivity ($<1\,\mathrm{dB}$ at $18\,\mathrm{kHz}$ according to the manufacturer) and also in sensitivity. This reduces systematic differences due to variations of the orientation of the instruments in the holes for different measurements.

3. We perform our measurements for a large number of distances from $5\,\mathrm{m}$ to $90\,\mathrm{m}$. This allows for the determination of the attenuation with a large lever arm of multiples of the attenuation lengths as well as the suppression of local glacial effects like cracks or reflections.

4. We include multiple measurements for the same distance but different locations on and depths in the glacier for the estimation of systematic uncertainties related to local properties of the glacier and reflections.

5. We include repeated measurements using the same holes that have been used a few days earlier, or of changed depth below the surface, to include uncertainties related to changing hole properties and thus acoustic coupling to the ice.

6. In each measurement, emitters and receivers are covered by a column of melted water at the bottom of the holes. The water interface improves the reproducibility of the coupling of the transducers to the ice.

7. We have developed a dedicated electronic setup for this measurement and tested it in the laboratory. The setup produces long signals of sine waves that are thus well defined in frequency. An appropriate time window of the registered sine-burst signals rejects transient ring-in phases until the receiver oscillates in phase as well as phases of electro-magnetic interferences.

8. In order to match the dynamic range for different distances to our setup, the amplitude of the sender can be changed. The emitted acoustic power is monitored in our setup for each measurement and differences are corrected for in the analysis by normalizing to the amplitude of the emitted signal. This approach also corrects for a possible long-term variation of the electronic setup in terms of gain. The validity of this normalization is verified *in situ* by measurements of different amplitude.

9. We perform the analysis very carefully by estimating and subtracting noise, identifying systematic uncertainties and a robust error propagation using advanced bootstrapping techniques.



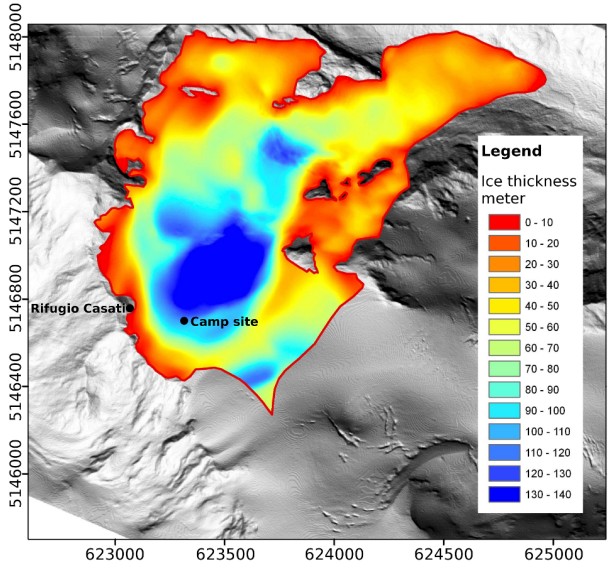

**Figure 1.** Extend and thickness map of the Langenferner glacier based on a modified figure in Stocker-Waldhuber (2010). The Casati hut and camp site of the field test are indicated. Coordinates are in UTM coordinates with east on the x-axis and north on the y-axis

## 2 The Measurement Setup

### 2.1 The Langenferner Site

The Langenferner is a high altitude glacier in the Ortler-Alps in Italy, that extends from its highest point at $3370\,\mathrm{m}$ a.s.l. to the lowest point at $2711\,\mathrm{m}$ a.s.l. at the terminus. Galos (2017) reports a covered area of of about $1.6\,\mathrm{km^2}$ (in 2013) and an

estimated volume of $0.08\,\mathrm{km^3}$ (in 2010).

    The site of the field test was located in the upper part of the glacier at about $3260\,\mathrm{m}$ a.s.l. close to the Rifugio Casati ($46.46\,^\circ$N|$10.60\,^\circ$E), see Fig. 1. The depth of the glacier in the region of the test site was estimated $90\,\mathrm{m}$ to $100\,\mathrm{m}$ in 2010 (Stocker-Waldhuber, 2010). Based on studies of the mass balance by Galos (2017), the site is part of the ablation zone and the depth was reduced by at least $7\,\mathrm{m}$ since 2010. During the field campaign, the glacier was not covered by snow and the ice

could be accessed directly.

    The instrumentation was deployed in the glacier by holes prepared with a $12\,\mathrm{cm}$ diameter melting probe that was developed within the EnEx initiative (Heinen et al., 2017). The layout of the holes at the test site is shown in Fig. 2, their coordinates and depths are detailed in Table 1. The figure shows that the test site includes complex ice structures though the main axis has been largely parallel to the largest visible cracks at the surface.

Inside the holes we have measured temperatures close to $0\,^\circ$C and the glacier appears largely tempered. However, we have observed over night that water surface of holes refroze and in some cases the acoustic transducers froze to the wall of the holes. Therefore domains in the bulk ice of slightly lower temperature cannot be excluded.





**Table 1.** Measurement holes. Coordinates are given in the UTM coordinate system (notation east|north|up) relative to hole 1 that is located at *(32U:623382.63|5146718.58|3281.84)*.

| # | Pos.<br>[m] | Coordinates<br>[m] | Depth<br>[m] |
|---|---|---|---|
| 1 | 0 | 0.00\|0.00\|0.00 | 2.6 |
| 2 | 5 | -5.02 \| -0.25 \| -0.36, | 1.8 |
| 3 | 10 | -10.09 \| -0.50 \|-0.81 | 2.1 |
| 4 | 30 | -30.27\| -4.20 \| -0.85 | 2.5 , 6[a] |
| 5 | 50 | -50.18 \|-2.75 \| -1.19 | 2.7 |
| 6 | 70 | -70.95 \| -0.91 \| -1.05 | 2.6 |
| 7 | 90 | -90.78 \| 0.47 \| -0.64 | 2.5 |

[a]changed $27^{th}$ August

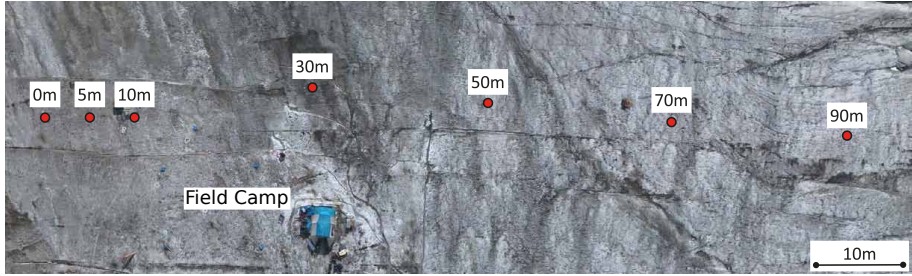

**Figure 2.** Aerial view of the measurement site with the location of the measurement holes. Modified photo from Markus Bobbe, TU Braunschweig.

## 2.2 Instrumentation and setup

The schematic overview of the measurement setup is shown in Fig. 3. Two spherical, $4.25\,\mathrm{inch}$, acoustic transducers of type ITC-1001 from *International Transducer Cooperation* are used for sending and receiving the signals. This type of transducer provides a high power broadband acoustic omni-directional emissivity from $2\,\mathrm{kHz}$ to $38\,\mathrm{kHz}$ and equally good receiving properties. These transducers are connected to the setup using coax cables and are lowered into the prepared and water-filled holes. All other equipment is contained in a secured metal box on the glacier to shield it from the outdoor environment. In each measurement the used transducer are not inter-changed for emitting and receiving the acoustic signals.

The setup is controlled through Ethernet connections by a notebook running LabVIEW. Signals are generated with a function generator (*Rigol DG5072*), amplified with a power amplifier (*Monacor PA-4040*) and sent to the emitter. The function generator also triggers the data acquisition that is done with a digital oscilloscope (*Tektronix DPO4034*). The signal of the acoustic receiver is amplified and synchronously recorded with this oscilloscope with a sampling rate of $1\,\mathrm{MHz}$. Because of the large





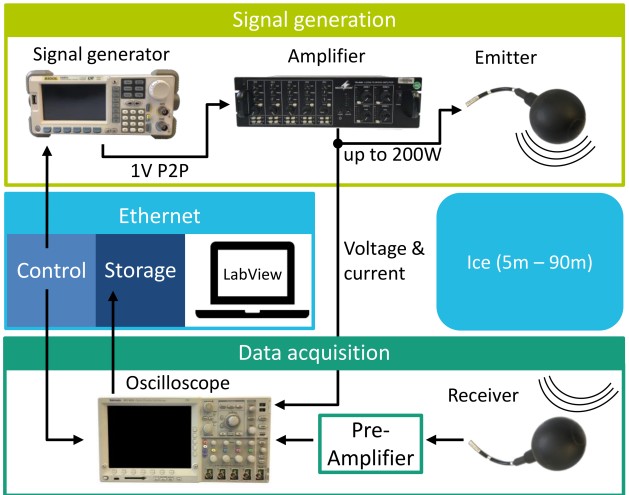

**Figure 3.** Schematics of the instrument setup.

difference of probed distances the electrical amplitude driving the emitter is dynamically adapted with peak-to-peak amplitudes ranging from $2\,\mathrm{V}$ to $500\,\mathrm{V}$. The LabView program automatically adjusts the dynamic range of the oscilloscope for maximum resolution of the received signal. Furthermore, we measure the power of the emitted signal during each measurement by monitoring the voltage and the current at the emitter input with a $1.1\,\Omega$ power resistor that is connected in series with the

emitter. In the data analysis the normalization of the received acoustic signals is corrected for the different emission power based on these recorded values.

### 2.3   Measurement procedures

Each measurement was carried out according to a strict procedure to ensure consistent data throughout the campaign. The spherical transducers were lowered to the bottom of the holes and were always covered by at least $30\,\mathrm{cm}$ of water. The main

attenuation measurement is based on repeated sine bursts of $50\,\mathrm{ms}$ duration. We scan for each pair of holes the frequency band of $2\,\mathrm{kHz}$ to $35\,\mathrm{kHz}$ in steps of $1\,\mathrm{kHz}$. To reduce ambient noise, the repeated burst signals of each frequency are averaged within the oscilloscope as indicated in Table 2. After one full frequency scan, the full procedure is repeated several times.

A measurement window of $100\,\mathrm{ms}$ was selected for the recording of data. This is substantially longer than the signal duration and allows recording $20\,\mathrm{ms}$ of ambient noise before a signal is emitted, and is sufficient to capture the complete signal including

a propagation delay of up to $30\,\mathrm{ms}$ that corresponds to a distance of more than $100\,\mathrm{m}$. The burst duration of $50\,\mathrm{ms}$ results in a minimum of 100 oscillations for the lowest frequency. This ensures a sufficiently long stable phase of forced resonance. By appropriate windowing during the offline analysis, phases of unstable amplitudes at the start and end of the burst are omitted. Similarly, phases of electromagnetic interferences are excluded from the analyzed time-windows, as described below.

In addition to these sine bursts, we have regularly recorded *logarithmic chirps* of $3\,\mathrm{ms}$, $5\,\mathrm{ms}$ and $10\,\mathrm{ms}$ duration within

frequency ranges between $0.5\,\mathrm{kHz}$ to $42.5\,\mathrm{kHz}$ as well as 11 bit Barker codes of $10\,\mathrm{kHz}$ and $20\,\mathrm{kHz}$ carrier frequency with





**Table 2.** Measurement runs

| # | Date | Dist. [m] | Holes | Avg. | Rep. | Dur. [hh:mm] |
|---|---|---|---|---|---|---|
| 6 | 23.08 | 60 | 6 → 3 | 512 | 3 | 04:33 |
| 7[a] | | 70 | 6 → 1 | 512 | 7 | 11:31 |
| 8[b] | | 10 | 1 → 3 | 512 | 1 | 00:35 |
| 9 | 24.08 | 10 | 1 → 3 | 128 | 4 | 01:53 |
| 10 | | 10 | 3 → 1 | 128 | 4 | 01:58 |
| 11[a] | | 50 | 5 → 1 | 128 | 35 | 17:08 |
| 12 | | 40 | 5 → 3 | 128 | 4 | 01:51 |
| 13 | 25.08 | 20 | 4 → 3 | 128 | 4 | 01:58 |
| 14 | | 30 | 4 → 1 | 128 | 4 | 01:51 |
| 15[c] | | 90 | 7 → 1 | 521 | 2 | 02:09 |
| 16 | | 20 | 5 → 4 | 128 | 4 | 01:52 |
| 17 | 26.08 | 40 | 6 → 4 | 128 | 4 | 01:52 |
| 18[d] | | 60 | 7 → 4 | 128 | 2 | 01:01 |
| 19 [a,c] | | 90 | 7 → 1 | 512 | 13 | 15:33 |
| 20 | | 40 | 7 → 5 | 128 | 4 | 01:52 |
| 21 | 27.08 | 20 | 6 → 5 | 128 | 4 | 01:51 |
| 22 | | 5 | 2 → 1 | 32 | 4 | 00:49 |
| 24[a] | | 60 | 6 → 3 | 512 | 9 | 15:18 |
| 25[e] | | 20 | 4 → 3 | 32 | 6 | 02:02 |
| 26 | 28.08 | 30 | 4 → 1 | 32 | 4 | 01:08 |
| 27 | | 25 | 4 → 2 | 32 | 1 | 00:26 |

[a]During night, [b]100% sending power, sine-bursts 2 kHz to 5 kHz, 25 kHz to 35 kHz only, [c]Sine-bursts 2 kHz to 25 kHz only, [d]Signal generator switched off, [e]Hole 4 deepened to 6 m

four oscillations per bit (Barker, 1953). These signals are used to determine the speed of sound. The chirps are also used for a second attenuation measurement with independent data.

An overview on the measurement runs that are used for the further data analysis is given in Table 2. Test runs and runs with data failures have been excluded from the list.





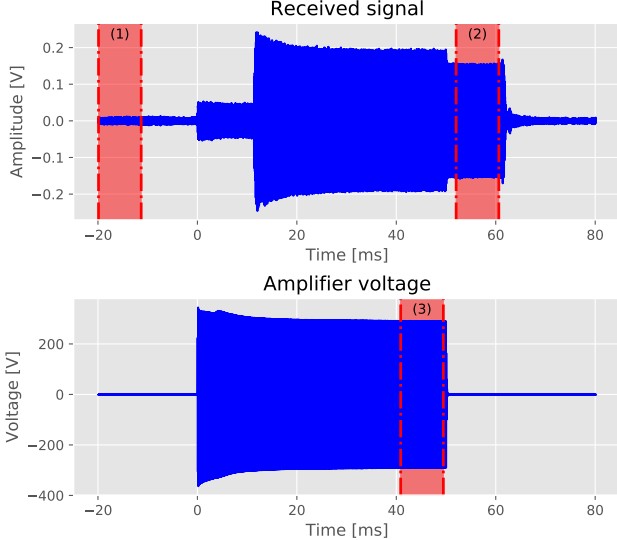

**Figure 4.** Waveform from measurement series 12 at 12 kHz (top) and the synchronously measured sending amplitude (bottom). The indicated windows 1 to 3 are relevant for the data analysis and are discussed in the text.

## 2.4 Waveform processing and amplitude extraction

Figure 4 shows as example a recorded waveform from the measurement series 12 for a 12 kHz burst at 40 m distance and the synchronously recorded signal that drives the emitter.

The recorded waveform features several characteristic properties that are explained in the following. From −20 ms to 0 ms
pure noise is recorded. Starting with the signal at 0 ms, we observe an cross-talk from electromagnetic interference in the received signal, that is identified due to its lack of propagation delay. After a delay of about 10 ms the acoustic signal sets in and is interfering with the cross-talk signal. Because the electromagnetic and acoustic signal have a constant relation in relative phase, the superposition is coherent. After 50 ms the sine-burst is switched off and immediately the interference in the received signal disappears. The now clean acoustic signal continues for the propagation delay up to about 60 ms, where it stops and the
receiver rings down.

### 2.4.1 Selection of analysis time windows in the waveforms

The electro-magnetic interference is caused by the high-power audio amplifier and the sensitive oscilloscope being packed very tightly in the metal box on the glacier. In the field we have verified by unplugging the emission cables that the cross-talk happens locally in the metal box and not at the receiving transducer. The amplitude of the cross-talk has been found to be
proportional to the sending amplitude. Note, that the frequency of the electromagnetic and the acoustic signal are the same for each measurement but the relative phase varies due to different propagation delays for different measurement. As result, we have observed both constructive as well as destructive interference between the two signals in the data. For the data analysis





we therefore use only acoustic data without interference. This can be easily accomplished, because for hole distances $d < 15\,\mathrm{m}$ sending amplitudes are small and received acoustic amplitudes are so large that the cross-talk can be neglected. At larger distances where the sending signal and corresponding cross-talk signal becomes larger, the propagation delay of the acoustic signal allows for a proper separation in time.

The selected windows are displayed in the example shown in Fig. 4. For the data processing we have selected for each measurement a window, (2) in Fig. 4, that contains the acoustic signal but no electromagnetic interference. Two windows of the same size are used to determine the noise in the causally unrelated region before the signal, (1) in Fig. 4, and, corrected for the propagation delay, in the recorded sending signal to determine the normalization of the sending signal, (3) in Fig. 4.

For distances $d < 15\,\mathrm{m}$, where the electromagnetic interference is negligible, we chose a signal window which is $20\,\mathrm{ms}$ delayed with respect to the start of the acoustic signal (to avoid ring-in effects) and a width of $19\,\mathrm{ms}$. For larger distances, the window starts with a margin of $2\,\mathrm{ms}$ after the end of the $50\,\mathrm{ms}$ long emission burst. The duration of the window depends on the distance assuming a propagation velocity of $3.6\,\mathrm{m\,ms^{-1}}$ minus a margin of $0.5\,\mathrm{ms}$. For distances of $80\,\mathrm{m}$ and above, the window width is limited to of $19\,\mathrm{ms}$. The proper adjustment of these windows has been applied for each measurement by an automated procedure but has been also visually verified during the analysis.

### 2.4.2 Fourier transformation

In the next step the data in each of the three time windows is Fourier transformed.

Though the three windows are already matched to the same width, they are further optimized with respect to the frequency of the respective sine burst such that exactly $N$ complete periods are inside the window, preventing spectral leakage due to incomplete periods. Furthermore, from the ratio of the signal- and sampling-frequencies the optimum number of data points fitting into this window is estimated. All signal windows are shortened accordingly. The shortening amounts to a maximum $0.5\,\mathrm{ms}$ for the $2\,\mathrm{kHz}$ signal.

Prior to the Fourier transformation, each signal window is multiplied with a Blackman window to further reduce boundary effects and spectral leakage. Since only the amplitude is of interest for the analysis, the absolute values of the Fourier transformation coefficients are taken, discarding the phase information.

The result of the transform is shown exemplary in Fig. 5 for the largest measured distance of $90\,\mathrm{m}$. The signal clearly exceeds the noise level with a SNR of about 10:1. The noise estimate in the noise window matches the noise-level for the signal window reasonably well. However, a precise prediction based on the different time window cannot be expected because of transient noise fluctuations.

### 2.4.3 Noise reduction by spectral subtraction

During the measurements we have observed that the noise level strongly varies with the time of day, i.e. the human activity on the glacier. Therefore the noise is subtracted from the signal Fourier spectrum for each measurement repetition $i$ individually. In order to avoid fluctuations, we average the values of the noise floor in a window $\pm 0.5\,\mathrm{kHz}$ around the respective target





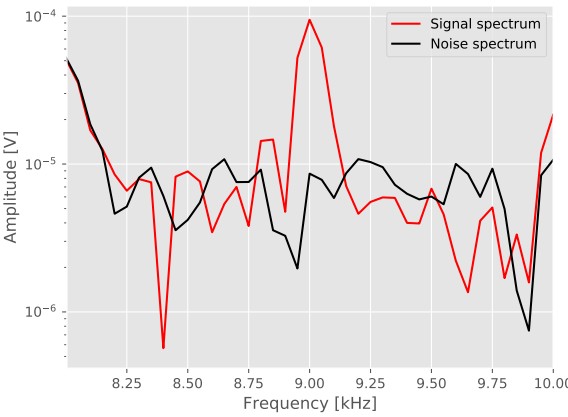

**Figure 5.** Frequency spectra for noise and signal windows for a burst measurement during series 19 at $9\,\mathrm{kHz}$.

frequency. The subtraction is performed quadratically $S_i(f) = \sqrt{Y_i^2(f) - \overline{N_i}^2}$, where $Y_i$ is the measured signal and $\overline{N_i}$ is the frequency averaged noise for the repetition i. This is based on the assumption, that the noise is uncorrelated in the time-domain.

We find generally a good SNR for all measurements and the noise subtraction is a rather small correction in most cases. Only for one waveform $Y_i^2(f) < \overline{N_i}^2$ was found, probably due to a strong transient signal overlapping with the measurement. This waveform from measurement series 7 over $70\,\mathrm{m}$ at $29\,\mathrm{kHz}$ has been discarded from the analysis.

Besides the subtraction of noise, the measured noise-level serves as uncertainty estimate of the measured signal $S_i$ and and we have used $\sigma_{S_i} = \overline{N_i}$.

### 2.4.4 Normalization to the emission power

Synchronously to the measured acoustic data, the sender's voltage $V$ and current $I$ are measured and stored as waveforms as shown in Fig. 4. These waveforms are Fourier transformed as well and the peak sending power $P_i = V \cdot A$ is determined by the multiplied coefficients of the target frequency. The normalized signal amplitude is given by $\hat{S}_i = S_i / \sqrt{\frac{P_i}{2}}$, where the factor $\sqrt{2}$ corrects the peak power to the effective sending power. The uncertainty $\sigma_{S_i}$ is multiplied with the same factor.

In the measurement series 8 and 9 we have verified the correctness of this normalization, by performing the same measurement but changing the emission power by a factor 200 resulting in highly different amplitudes, once close to the detection threshold and once close to saturation. The normalized amplidudes are found fully consistent.

### 2.4.5 Data averaging

The amplitude extraction is repeated for each repetition within one series, see Table 2. We have observed, that particularly during long measurement series both extracted signal and noise level can vary significantly between measurements. Therefore we calculate for each series $n$ the error weighted mean of all N repetitions $S_n = \frac{\sum_{i=1}^{N} S_i / \sigma_i^2}{\sum_{i=1}^{N} 1/\sigma_i^2}$ and the corresponding error $\sigma_n = $



$\sqrt{\frac{1}{\sum_{i=1}^{N} 1/\sigma_i^2}}$. Deviations from these averages are assumed to be caused by systematic uncertainties and will be investigated in the following.

## 2.5 Stability of data in time

For the estimation of the total uncertainty of each measurement, we have to take into account several effects

1. Changes of the extracted signal for different repetitions during long measurement series result in an error $\sigma_{S,i}$ of the averaged value in addition to the propagated errors $\sigma_n$.

2. Differences of the extracted signal for repeated measurements in the same hole but different dates n and m indicate systematic variations of the glacial conditions during the measurement campaign. This additional uncertainty is named $\sigma_{S_{n,m}}$.

3. Differences of the extracted signal ratio for pairs of two holes at the same distance but different positions on the glacier and dates of the measurement indicate the uncertainty related to the local position on the glacier. This additional uncertainty is called $\sigma_{S_n,S_m}$.

The total uncertainty for each signal $S_i$ is then given by

$$\sigma = \sqrt{\sigma_n^2 + \sigma_{S,i}^2 + \sigma_{S_{n,m}}^2 + \sigma_{S_n,S_m}^2}, \tag{1}$$

where each uncertainty includes the additional uncertainty related to the respective effect.

### 2.5.1 Observed changes during measurement series

The repeated measurements during long measurement series allow for the investigation of systematic changes of the measured amplitudes over time. Figure 6 and 7 show as examples the results from two measurement series of more than $10\,\mathrm{h}$ run time and a large number of repetitions. While the results in the first example are stable within their uncertainties, the second example

shows a systematic variation exceeding the assumed errors.

The origin of this effect remains unclear. However, we can exclude instrumental effects because all diagnostic data indicates stable operation for these runs. Therefore, we suspect variations of the glacier itself, i.e. spontaneous relaxation of cracks, refreeze of melting water within cracks during night as well as changes of the geometry of the melted holes including the water-level and the acoustic coupling of the sensor and emitters to the bulk ice.

In order to account for such changes in the error budget, we calculate the standard deviation $std(S_i)$. If this error is in excess of the previously estimated error from the mean of the repeated measurements it is added to the total error by $\sigma_{S,i}^2 = \sup(0, std(S_i)^2 - \sigma_n^2)$.





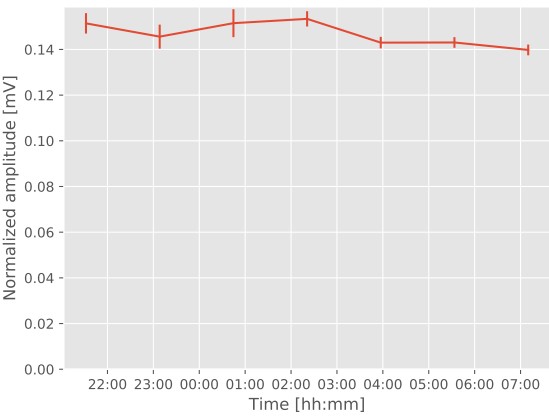

**Figure 6.** Measured amplitude for repeated measurements within series 7, 19 kHz

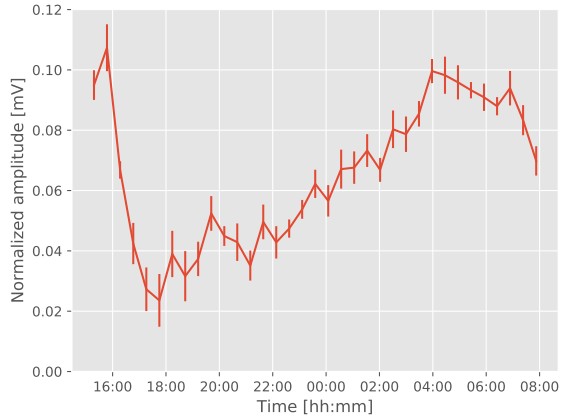

**Figure 7.** Measured amplitude for repeated measurements within series 11, 27 kHz.

### 2.5.2 Reproducibility of measurements for repeated series

To assess the reproducibility of full measurement series, three pairs of measurement series were taken between the same holes: 9 and 10 (10 m, directly consecutive), 6 and 24 (60 m, 4 days apart) and 15 and 19 (90 m, 1 day apart). In between, the setups had been removed from their holes and then reinstalled.

5    Figure 8 shows the amplitude plotted against the frequency for all six measurement series. Overall, all three pairs show a reasonably good consistency of the amplitude and shape of the curve within the estimated uncertainties. However, also significant differences can be seen, e.g. for measurement series 6 and 24.





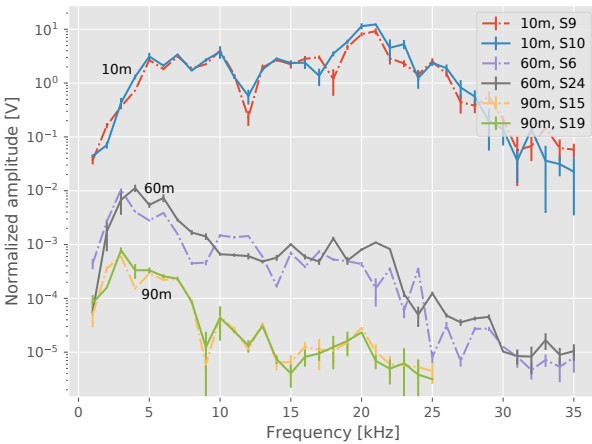

**Figure 8.** Amplitudes of measurement series 9 and 10 (10 m), 6 and 24 (60 m) and 15 and 19 (90 m).

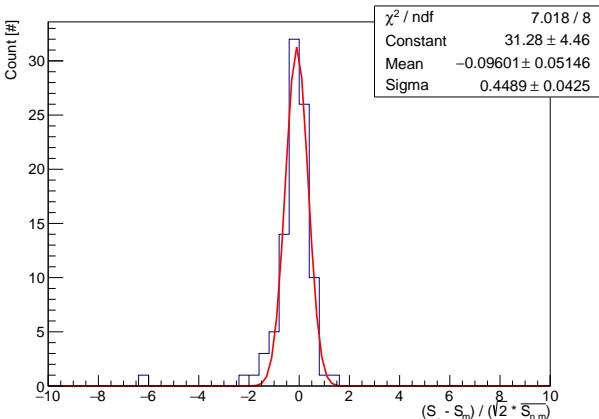

**Figure 9.** Histogram of the relative variations between repeated measurements of the same hole pairs for all frequencies

In order to account for the variations in reproducibility we have investigated all measured relative differences $s_{nm} = (S_n - S_m)/(\sqrt{2} \cdot \overline{S_{n,m}})$. We find no dependency on the frequency and use the standard deviation $std(s_{nm}) = 0.45$ of this distribution (see Fig. 9) to account for the systematic uncertainty of time variations at fixed locations on the glacier $\sigma_{S_{n,m}} = 0.45 \cdot S_i$.

### 2.5.3 Systematic differences related to different pairs of holes

5    Figure 10 shows as an example the measured amplitudes as a function of the hole distance for 16 kHz sine bursts. The semi-logarithmic plot displays a roughly linear dependency of amplitude and distance as expected. However, variations in amplitude exceeding the uncertainties of the individual measurements are visible at distances 20 m, 40 m and 60 m, see Table 2 for details





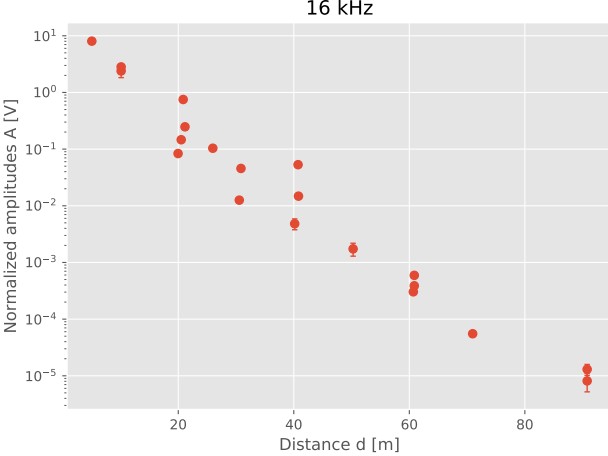

**Figure 10.** Normalized amplitudes $16\,\text{kHz}$ sine bursts. Variations in measured amplitudes for measurements of different hole pairs at $20\,\text{m}$, $40\,\text{m}$ and $60\,\text{m}$ are indicated.

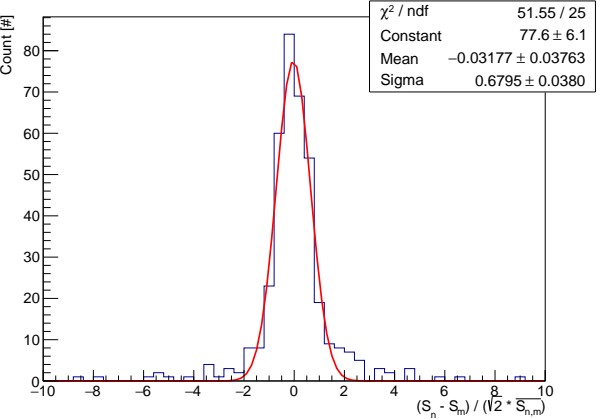

**Figure 11.** Histogram of the relative difference between measurements of hole pairs of the same distance for all frequencies

on the measurement series. Note, that this figure also displays the variations of repeated measurements of the same hole-pairs ($10\,\text{m}$, $60\,\text{m}$ and $90\,\text{m}$) that are discussed in the previous section.

In order to estimate the uncertainty due to the propagation of signals through different ice masses, we have again investigated all relative differences of measured amplitudes of different hole pairs $(S_n - S_m)/(\sqrt{2} \cdot \overline{S_{n,m}})$ and estimated the standard

5  deviation $std(s_n, s_m) = 0.68$. As this variation includes also the variation due to the time dependency, observed when using same holes, we subtract this respective uncertainty as estimated above $\sigma^2_{S_n, S_m} = std(s_n, s_m)^2 - \sigma^2_{\overline{S_{n,m}}} = (0.68^2 - 0.45^2) \cdot S_i^2 = 0.51^2 \cdot S_i^2$.





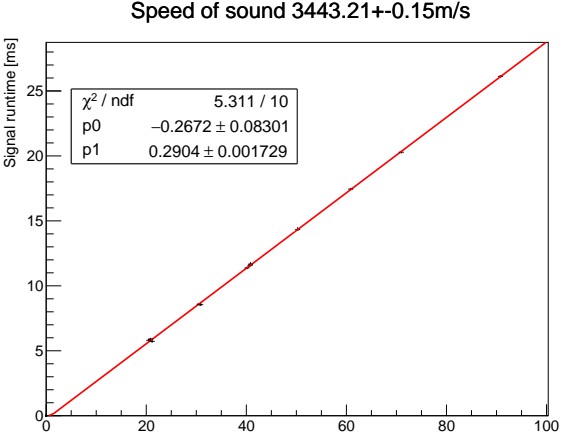

**Figure 12.** Measured propagation delay for $5\,\mathrm{ms}$ chirp signals

**Table 3.** Measurement of the propagation speed of sound $v_{prop}$.

|  | $v_{prop}$ | $\chi^2/n_{dof}$ |
|---|---|---|
|  | m/s |  |
| chirp ($3\,\mathrm{ms}$) | $3443.0 \pm 0.2$ | $5.31/10$ |
| chirp ($5\,\mathrm{ms}$) | $3443.2 \pm 0.2$ | $5.31/10$ |
| chirp ($10\,\mathrm{ms}$) | $3447.9 \pm 0.2$ | $0.70/10$ |
| barker | $3477.0 \pm 0.1$ | $116.9/10$ |

## 2.6 Speed of Sound Measurement

An important verification of the *in situ* performance of the setup is the measurement of the speed of sound. For this measurement, we use the transmitted chirp and barker signals and estimate the propagation delay by the maximum correlation of emitted and received signals.

5    The used signals of $3\,\mathrm{ms}$ to $10\,\mathrm{ms}$ are shorter than the typical propagation delay of the acoustic wave. To avoid any influence of the electromagnetic induced signals, only measurements of distances larger than $10.8\,\mathrm{m}$ ($3\,\mathrm{ms}$), $18.0\,\mathrm{m}$ ($5\,\mathrm{ms}$) and $36\,\mathrm{m}$ ($10\,\mathrm{ms}$) are used as the signal emission is terminated before the acoustic signal reaches the receiver. The time-window of the electro-magnetic interference is excluded from the analysis.

The propagation delay is calculated by correlating for each measurement the recorded emitter voltage with the received
10    signal with a variable time offset. The time offset of maximum correlation determines the signal propagation time. The median from all repetitions of the same measurement is taken as well as the difference of the $15.85\,\%$ and $84.15\,\%$ quantiles for an estimate of the error.





The result of the measured propagation delay is summarized in Table 3 and shown in Fig. 12 for the example of $5\,\mathrm{ms}$ chirps. We observe a good linear behavior of the propagation delay with distance. From the chirp signals, a combined speed of sound of $(3444.7 \pm 1.6)\,\mathrm{m/s}$ is observed. The dominant systematic uncertainty on the absolute value of the speed of sound is related to the determination of the hole locations. The location of each hole has been measured with a GPS probe that showed a drift

of about $80\,\mathrm{cm}$ during the procedure. This drift corresponds to an uncertainty of about $30\,\mathrm{m/s}$.

The results for different chirps signals are, however, fully correlated with respect to this uncertainties and can be directly compared. The results of the $3\,\mathrm{ms}$ and $5\,\mathrm{ms}$ chirps are consistent with each other within their estimated fit-errors. The speed of sound derived from the $10\,\mathrm{ms}$ chirps deviates by about $5\,\mathrm{m/s}$ from those, and is thus not consistent within the errors that have been estimated from the fit. The barker signals show substantially stronger fluctuations in the propagation time which is

also reflected by a large $\chi^2$ value. The observed speed of sound deviates by $30\,\mathrm{m/s}$ from the results of the chirps. The barker signals are thus not taken into account in the further analysis.

We conclude, that the measured propagation delay sufficiently verifies the stability of the measurement setup. However, it also indicate not fully understood systematic uncertainties related to barker signals.

Our measured value of the speed of sound is smaller than $3880\,\mathrm{m/s}$ as measured for deep antarctic ice but larger than the

observations for firn ice (Abbasi et al., 2010). It is only slightly smaller than a previous measurement near the surface of alpine glaciers and antarctic glaciers with about $3660\,\mathrm{m/s}$ to $3700\,\mathrm{m/s}$ and $3500\,\mathrm{m/s}$ respectively (Helbing et al., 2016). However, there it was also observed that the propagation delay strongly depends on the direction and depth in the ice with variations up to $\pm 10\,\%$. This indicates a strong dependency on the structure of the ice and the morphology of the glacier. When taking into account these systematic uncertainties, we consider our observed value as a reasonably good confirmation of our measurement

procedures.

## 2.7 Attenuation using Chirp signals

The measured chirp signals can also be used to measure the attenuation of sound. For this, we have adopted a procedure that is mostly identical to the above described procedure in terms of estimation of uncertainties. Unlike the above procedure, the total received chirp signal as well as a noise window are Fourier transformed and the amplitude at the respective frequency is used

after noise subtraction. The Fourier transformation is recalculated for each frequency with a window length adjusted to this frequency in order to minimize spectral leakage. In comparison to the sine-burst measurement we do not measure a frequency clean signal and e.g. transient ringing of the receiver cannot be fully excluded from the measurement as easy. Furthermore, an uncertainty in the frequency dependency of the speed of sound and surface reflections may result in an uncertainty due to the dispersion of received signal. As the analysis of this data is thus less robust against these uncontrolled uncertainties we use this

independent data-set for a second measurement confirming our main result that is based on the sine-bursts.

As detailed for the measurement for the speed of sound, electromagnetic interference is no problem in case of chirps. Since the emission is terminated quickly, an overlap of the interference and the received acoustic signal happens only for short distances below $10.8\,\mathrm{m}$ ($3\,\mathrm{ms}$), $18.0\,\mathrm{m}$ ($5\,\mathrm{ms}$) and $36\,\mathrm{m}$ ($10\,\mathrm{ms}$) with an speed of sound of $3600\,\mathrm{m/s}$. As for the sine bursts, for all measurements up to distances of $20\,\mathrm{m}$ the electromagnetic interference is negligible due to the combination of high received





**Table 4.** Estimated values for the relative systematic uncertainties $\sigma_{S_{n,m}}$ and $\sigma_{S_n,S_m}$ for the chirp measurements. For comparison, also the results from the sine burst measurements are listed.

| Signal | $\sigma_{S_{n,m}}$ | $\sigma_{S_n,S_m}$ |
|---|---|---|
| 3 ms chirps | 0.39 | 0.34 |
| 5 ms chirps | 0.41 | 0.40 |
| 10 ms chirps | 0.32 | 0.51 |
| all chirps | 0.38 | 0.48 |
| sine | 0.45 | 0.51 |

acoustic amplitude and low sending power. Thus we have excluded only the $10\,\mathrm{ms}$ chirp measurement series 14, 26 and 27 which are in the range of $20\,\mathrm{m}$ to $35\,\mathrm{m}$.

The relative systematic uncertainties $\sigma_{S_{n,m}}$ and $\sigma_{S_n,S_m}$ are listed in table 4 for the three chirp durations separately and for the combination of all chirps.

When fitting for the attenuation lengths (see below), we observe no systematic differences for chirps of different duration. Therefore we combine the full data set of all chirps, without distinction by duration for the final result.

## 3    Result of the Attenuation measurement

The acoustic attenuation is measured by fitting the determined sound amplitudes as a function of distance $d$ for each frequency with the function

$$A(d) = \frac{A_0}{d} \cdot e^{-\frac{d}{\lambda_{att}}} + N \ . \tag{2}$$

Free parameters of the fit are the amplitude normalization $A_0$, the attenuation length $\lambda_{att}$ and the amplitude of the noise floor $N$. Note, that this function ignores the effect of surface reflections.

The error of each data point includes the estimations of the individually measured signal to noise ratio but also accounts for systematic variations that we have observed in the data as described above. For each frequency $f$ and measurement series $n$,

this results in the amplitude and error

$$A(d) = S_n \pm \sqrt{\sigma_n^2 + \sigma_{S,i}^2 + S_n^2 \cdot (0.45^2 + 0.51^2)} \ . \tag{3}$$

In order to increase the robustness of the analysis we include all 20 measured data series but repeat the fit multiple times with a subset of these points. Each of this subset contains 20 random data points where each point can appear multiple times but the total number of points remains constant. This is a resampling technique called *bootstrapping* which provides a rather

robust estimate of the uncertainties driven by the fluctuations in the data, i.e. outliers (Narsky and Porter, 2013).

We repeat this bootstrapping 1000 times for each frequency and perform the fit. For a robust estimate against stochastic outliers we then use the median ($50\,\%$ quantile) as well the $15.85\,\%$ and $84.15\,\%$ quantiles from the results of the 1000 fits as





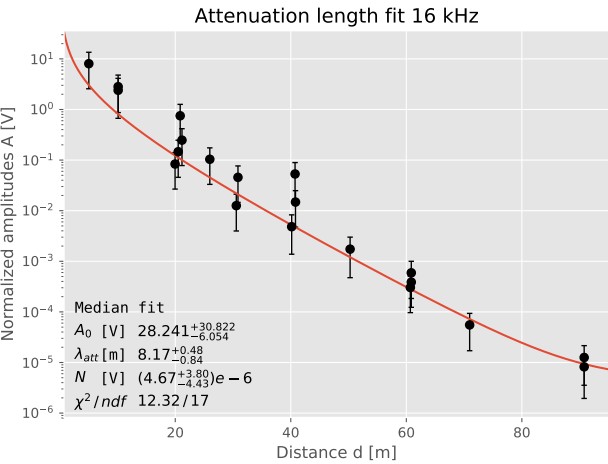

**Figure 13.** Fit of attenuation length $\lambda_{att}$ for $16\,\mathrm{kHz}$. The line and the $\chi^2$ is calculated with the median parameters from the 1000 bootstrap estimates.

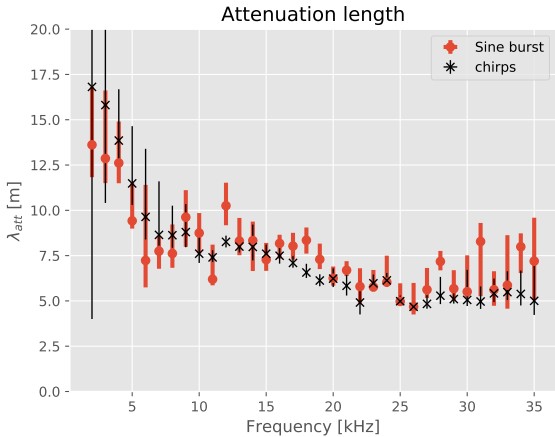

**Figure 14.** Attenuation lengths for all frequencies. Shown are the results based on sine-bursts (red bullets) as well as chirps (black stars)

the asymmetric error of the fit results. An example for $16\,\mathrm{kHz}$ is shown in Fig. 13. The averaged fit agrees well with the data points within uncertainties and is not driven by outliers. All fitted parameters with their estimated uncertainty are listed for each frequency in Table 5. The fitted attenuation length versus frequency is shown in Fig. 14.

The resulting uncertainties of the attenuation length are typically $20\,\%$ and include systematic uncertainties as described above. Note also that the measurement of each frequency is based on independent data. The values of the $\chi^2$ represent a $\chi^2$-test of all data points with respect to the average fit. The number of degrees of freedom slightly varies, because for the lowest and largest frequencies data has not been taken for the largest distances as the observed signal was too weak. The values of the $\chi^2$





**Table 5.** Results of the fitting for all frequencies. Left values for sine signals, right for chirps.

| $f$ [kHz] | $A_0$ [V] | $\lambda_{att}$ [m] | $N$ [V·$10^{-5}$] | $\chi^2/ndf$ | $A_0$ [V] | $\lambda_{att}$ [m] | $N$ [V·$10^{-5}$] | $\chi^2/ndf$ |
|---|---|---|---|---|---|---|---|---|
| 2 | $1.2^{+0.4}_{-0.5}$ | $13.6^{+3.2}_{-1.8}$ | $17.06^{+6.63}_{-7.58}$ | 10.7/16 | $3.2^{+13.9}_{-3.0}$ | $16.8^{+33.2}_{-12.8}$ | $29.35^{+38.75}_{-24.26}$ | 189.1/16 |
| 3 | $5.0^{+0.0}_{-1.3}$ | $12.9^{+3.8}_{-1.4}$ | $35.85^{+12.83}_{-29.02}$ | 16.6/16 | $5.1^{+9.8}_{-4.4}$ | $15.8^{+12.0}_{-5.4}$ | $43.67^{+18.33}_{-43.67}$ | 82.5/16 |
| 4 | $13.1^{+0.0}_{-5.8}$ | $12.6^{+2.3}_{-1.1}$ | $8.73^{+12.92}_{-8.73}$ | 12.9/16 | $7.0^{+3.2}_{-2.0}$ | $13.9^{+2.8}_{-1.4}$ | $12.42^{+5.38}_{-12.42}$ | 12.1/16 |
| 5 | $29.6^{+1.7}_{-10.9}$ | $9.4^{+2.1}_{-0.4}$ | $25.09^{+3.55}_{-11.14}$ | 10.7/16 | $13.6^{+6.5}_{-4.7}$ | $11.5^{+3.2}_{-1.2}$ | $17.10^{+3.56}_{-17.10}$ | 8.2/16 |
| 6 | $21.2^{+46.8}_{-12.2}$ | $7.2^{+4.2}_{-1.5}$ | $11.25^{+7.49}_{-10.48}$ | 23.3/17 | $21.8^{+11.9}_{-11.1}$ | $9.6^{+3.8}_{-1.2}$ | $16.56^{+4.73}_{-12.50}$ | 10.8/17 |
| 7 | $47.4^{+38.7}_{-16.2}$ | $7.7^{+0.8}_{-1.0}$ | $10.60^{+12.66}_{-7.46}$ | 13.7/17 | $41.6^{+11.4}_{-21.2}$ | $8.6^{+3.0}_{-0.6}$ | $12.65^{+2.09}_{-7.98}$ | 8.2/17 |
| 8 | $50.0^{+31.8}_{-31.2}$ | $7.6^{+1.6}_{-0.8}$ | $8.60^{+1.47}_{-1.63}$ | 11.9/17 | $35.5^{+17.6}_{-13.4}$ | $8.6^{+1.6}_{-0.7}$ | $6.66^{+1.77}_{-2.98}$ | 8.5/17 |
| 9 | $10.0^{+39.6}_{-4.2}$ | $9.6^{+1.5}_{-1.6}$ | $0.00^{+0.12}_{-0.00}$ | 20.6/17 | $27.6^{+22.0}_{-11.3}$ | $8.8^{+1.5}_{-0.8}$ | $5.26^{+1.92}_{-2.33}$ | 10.5/17 |
| 10 | $24.2^{+29.2}_{-8.7}$ | $8.7^{+1.1}_{-1.1}$ | $3.67^{+0.81}_{-1.48}$ | 11.3/17 | $58.5^{+29.9}_{-20.4}$ | $7.6^{+1.2}_{-0.5}$ | $6.08^{+1.49}_{-1.36}$ | 7.8/17 |
| 11 | $88.8^{+3.2}_{-74.4}$ | $6.2^{+1.9}_{-0.3}$ | $2.33^{+0.33}_{-0.60}$ | 15.2/17 | $85.8^{+14.2}_{-15.0}$ | $7.4^{+0.4}_{-0.3}$ | $4.39^{+0.82}_{-0.43}$ | 5.0/17 |
| 12 | $10.4^{+12.5}_{-4.6}$ | $10.3^{+1.3}_{-1.1}$ | $0.00^{+0.00}_{-0.00}$ | 13.1/17 | $50.2^{+10.8}_{-8.0}$ | $8.3^{+0.3}_{-0.3}$ | $2.70^{+0.45}_{-0.51}$ | 5.0/17 |
| 13 | $40.2^{+29.0}_{-20.6}$ | $8.3^{+1.3}_{-0.8}$ | $2.32^{+0.64}_{-1.09}$ | 5.6/17 | $67.3^{+10.1}_{-7.8}$ | $8.0^{+0.2}_{-0.3}$ | $3.50^{+0.79}_{-0.49}$ | 2.6/17 |
| 14 | $20.3^{+87.3}_{-10.5}$ | $8.3^{+1.0}_{-1.7}$ | $0.27^{+0.33}_{-0.27}$ | 16.1/17 | $45.8^{+46.6}_{-24.0}$ | $8.0^{+1.2}_{-0.7}$ | $4.17^{+1.32}_{-1.70}$ | 10.2/17 |
| 15 | $46.7^{+17.0}_{-24.3}$ | $7.3^{+0.9}_{-0.6}$ | $0.32^{+0.25}_{-0.32}$ | 13.5/17 | $78.0^{+32.4}_{-21.2}$ | $7.6^{+0.5}_{-0.4}$ | $4.47^{+1.27}_{-2.83}$ | 7.7/17 |
| 16 | $28.2^{+30.8}_{-6.1}$ | $8.2^{+0.5}_{-0.8}$ | $0.47^{+0.38}_{-0.44}$ | 12.3/17 | $93.1^{+27.6}_{-23.0}$ | $7.5^{+0.5}_{-0.4}$ | $5.50^{+3.97}_{-5.30}$ | 9.7/17 |
| 17 | $48.0^{+24.9}_{-17.8}$ | $8.0^{+0.7}_{-0.7}$ | $0.38^{+0.40}_{-0.38}$ | 7.0/17 | $144.1^{+24.8}_{-26.0}$ | $7.1^{+0.4}_{-0.3}$ | $11.56^{+1.84}_{-1.99}$ | 4.5/17 |
| 18 | $39.6^{+31.1}_{-10.1}$ | $8.3^{+0.7}_{-0.7}$ | $0.21^{+0.48}_{-0.21}$ | 11.6/17 | $264.9^{+45.1}_{-58.7}$ | $6.6^{+0.5}_{-0.3}$ | $20.43^{+6.28}_{-4.04}$ | 6.0/17 |
| 19 | $105.0^{+69.8}_{-46.0}$ | $7.3^{+0.9}_{-0.5}$ | $1.14^{+0.27}_{-0.47}$ | 6.4/17 | $449.4^{+104.1}_{-91.3}$ | $6.1^{+0.3}_{-0.3}$ | $29.90^{+11.42}_{-7.17}$ | 7.2/17 |
| 20 | $186.4^{+0.0}_{-74.7}$ | $6.2^{+0.6}_{-0.4}$ | $3.01^{+4.79}_{-0.67}$ | 15.9/17 | $425.9^{+226.8}_{-94.5}$ | $6.2^{+0.6}_{-0.5}$ | $46.03^{+88.14}_{-12.84}$ | 8.8/17 |
| 21 | $119.0^{+45.0}_{-36.6}$ | $6.7^{+0.5}_{-0.3}$ | $0.64^{+0.29}_{-0.19}$ | 12.5/17 | $601.9^{+345.5}_{-310.8}$ | $5.8^{+0.7}_{-0.5}$ | $37.76^{+15.70}_{-9.39}$ | 10.7/17 |
| 22 | $187.3^{+18.9}_{-155.6}$ | $5.8^{+1.0}_{-0.9}$ | $0.53^{+0.14}_{-0.11}$ | 38.4/17 | $640.4^{+774.4}_{-180.1}$ | $4.9^{+1.1}_{-0.7}$ | $36.47^{+16.09}_{-8.81}$ | 15.7/17 |
| 23 | $166.3^{+45.6}_{-113.4}$ | $5.8^{+0.9}_{-0.2}$ | $0.57^{+0.10}_{-0.11}$ | 8.6/17 | $328.8^{+124.8}_{-81.4}$ | $6.0^{+0.5}_{-0.3}$ | $23.37^{+3.85}_{-2.85}$ | 7.7/17 |
| 24 | $83.3^{+29.5}_{-68.9}$ | $6.0^{+1.5}_{-0.2}$ | $0.37^{+0.09}_{-0.16}$ | 9.2/17 | $234.4^{+99.7}_{-59.4}$ | $6.1^{+0.4}_{-0.4}$ | $19.60^{+2.92}_{-2.83}$ | 6.8/17 |
| 25 | $128.9^{+0.0}_{-103.7}$ | $5.0^{+1.0}_{-0.2}$ | $0.28^{+0.10}_{-0.2}$ | 10.2/17 | $417.7^{+60.7}_{-81.5}$ | $5.0^{+0.3}_{-0.1}$ | $19.82^{+3.55}_{-3.02}$ | 5.5/17 |
| 26 | $97.7^{+0.0}_{-78.7}$ | $4.7^{+1.3}_{-0.4}$ | $0.99^{+0.50}_{-0.45}$ | 20.2/15 | $444.9^{+67.8}_{-111.4}$ | $4.7^{+0.3}_{-0.1}$ | $22.50^{+5.90}_{-2.73}$ | 4.3/17 |
| 27 | $26.2^{+20.6}_{-17.9}$ | $5.6^{+1.2}_{-0.5}$ | $0.21^{+0.20}_{-0.21}$ | 8.5/15 | $208.8^{+59.3}_{-68.6}$ | $4.8^{+0.5}_{-0.3}$ | $18.59^{+6.96}_{-2.57}$ | 5.5/17 |
| 28 | $4.4^{+1.2}_{-0.6}$ | $7.2^{+0.6}_{-0.5}$ | $0.33^{+1.35}_{-0.33}$ | 12.7/15 | $104.2^{+33.9}_{-63.1}$ | $5.3^{+1.0}_{-0.5}$ | $27.93^{+5.05}_{-4.11}$ | 7.4/17 |
| 29 | $16.8^{+0.0}_{-11.2}$ | $5.7^{+1.0}_{-0.7}$ | $0.18^{+0.48}_{-0.18}$ | 20.7/15 | $83.9^{+17.5}_{-25.4}$ | $5.1^{+0.7}_{-0.3}$ | $28.86^{+8.51}_{-4.44}$ | 3.8/17 |
| 30 | $8.6^{+0.0}_{-6.5}$ | $5.5^{+2.0}_{-0.3}$ | $0.47^{+0.43}_{-0.32}$ | 8.0/15 | $68.5^{+17.5}_{-45.7}$ | $5.1^{+1.7}_{-0.3}$ | $28.99^{+53.60}_{-7.38}$ | 4.7/17 |
| 31 | $0.6^{+3.9}_{-0.2}$ | $8.3^{+1.0}_{-3.0}$ | $0.11^{+0.31}_{-0.11}$ | 11.8/15 | $56.5^{+18.8}_{-22.6}$ | $5.0^{+0.8}_{-0.4}$ | $29.04^{+15.99}_{-5.80}$ | 3.8/17 |
| 32 | $6.6^{+0.1}_{-5.2}$ | $5.6^{+1.0}_{-0.9}$ | $0.38^{+0.15}_{-0.13}$ | 43.3/15 | $41.8^{+24.5}_{-22.2}$ | $5.4^{+0.8}_{-0.5}$ | $35.62^{+9.26}_{-5.05}$ | 3.7/17 |
| 33 | $3.4^{+4.8}_{-3.0}$ | $5.9^{+2.8}_{-1.3}$ | $0.31^{+0.66}_{-0.20}$ | 19.9/15 | $48.4^{+21.8}_{-27.7}$ | $5.5^{+1.2}_{-0.4}$ | $35.09^{+9.66}_{-4.61}$ | 4.4/17 |
| 34 | $0.6^{+1.2}_{-0.2}$ | $8.0^{+0.7}_{-1.5}$ | $0.25^{+0.17}_{-0.19}$ | 8.7/15 | $48.0^{+38.5}_{-30.5}$ | $5.4^{+1.3}_{-0.6}$ | $36.50^{+9.62}_{-5.10}$ | 6.5/17 |
| 35 | $0.6^{+3.4}_{-0.3}$ | $7.2^{+2.4}_{-2.2}$ | $0.45^{+0.47}_{-0.25}$ | 12.5/15 | $74.5^{+46.0}_{-61.2}$ | $5.0^{+2.0}_{-0.8}$ | $43.64^{+10.05}_{-6.25}$ | 8.8/17 |

are found to be reasonable for all fits. Note also, that the fit values for the noise floor $N$ are for all fits in agreement with zero, thus verifying the noise reduction is working well and does not introduce a bias to the fit.



Also shown in the figure is the result of the chirp measurement. The attenuation that is obtained with this independent data set is found to be consistent with the sine-burst measurement in absolute and remarkably even structures of the frequency dependency. We interpret this as a good confirmation of the result.

Two systematic effects that are hard to control experimentally have to be addressed. That is first the coupling of the sound from and into the water-filled holes. In the holes standing waves are expected to build up at characteristic frequency which may modify the angular response. Secondly reflections from the surface will constitute a coherent wave that may interfere constructively or destructively with the received signal. Both effects are expected to vary strongly with distance, depth of holes and probed frequencies but will not constitute as an exponential-like distance dependence given the large lever-arm of performed measurements. No obvious contribution from these effects has been found neither in the raw waveform data nor in the frequency and distance dependency of measured amplitudes. The absence of strong surface reflections is in fact plausible because of the highly uneven and rough surface on scales of the wave-length that diminishes the coherence of reflected signals, see Fig. 2. The remaining contribution leading to fluctuations of individual data points are included in the estimation of systematic errors by repeated measurements. Any further impact of such fluctuations on the fit are further suppressed by the bootstrapping method. The validity of these assumptions is confirmed by the consistency of results of the chirp and the sine-burst measurements because both would be affected differently by these effects.

## 4 Discussion and Conclusions

In this paper we report the measurement of the acoustic attenuation length on the alpine glacier Langenferner in the frequency range from $2\,\mathrm{kHz}$ to $35\,\mathrm{kHz}$. The range of values are typically $5\,\mathrm{m}$ to $15\,\mathrm{m}$ with larger attenuation length for lower frequency. These values include a detailed investigation of systematic uncertainties and are based on two independent measurements using sine-bursts and chirp signals. The measured speed of sound is about $(3447 \pm 3)\,\mathrm{m/s}$.

Figure 15 shows a comparison of our result to that of Langleben (1969) obtained for sea ice. Despite of the large spread in the sea ice data, our result agrees well with that in the range $10\,\mathrm{kHz}$ to $25\,\mathrm{kHz}$. Above we find a smaller attenuation. Also compared to Westphal (1965) we find a result that is consistent in magnitude but we observe a weaker frequency dependency. In conclusion our data does not favor the model of Rayleigh scattering as dominant effect of the attenuation. According to Price (1993, 2006) the dominant effect of energy loss of acoustic waves in warm ice is grain boundary relaxation, i.e. sliding. This process has a weaker frequency dependency than scattering and depends on the texture of the ice and its grain size. For colder ice, this process is suppressed.

When comparing to the results for the alpine glaciers Pers and Moteratsch, reported in Helbing et al. (2016) we find an attenuation length that is shorter by approximately a factor 2 but remarkably a similar frequency dependency. The glacial environment and measurement strategies are quite similar, however, the origin of this difference is unclear. We note, that despite of these differences, the measured attenuation of sound is remarkably similar in scale for very different locations, e.g. sea-ice and different alpine glaciers when taking into accounts the large difference to deep antarctic ice. Even unpublished measurements by ourselves during a campaign on the non-tempered Canada glacier in Antarctica (Kowalski et al., 2016)



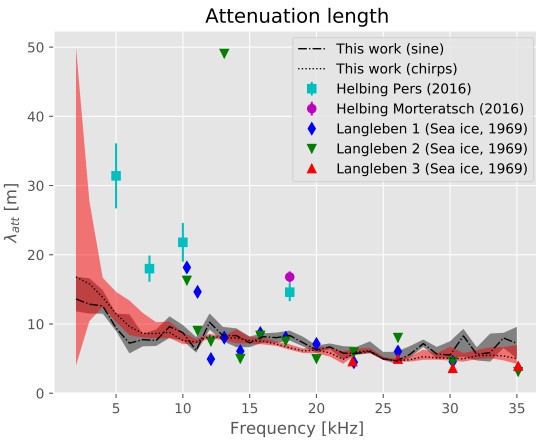

**Figure 15.** Comparison of our measurement to the results from Langleben (1969) for sea ice and Helbing et al. (2016). Shown are our results based on sine-bursts (dark grey band) as well as chirps (light red band) and the other reported results as data points

resulted in similar values. Further follow-up measurements on different glaciers would be required to confirm whether the effective attenuation of sound is a general property and whether it is related to the specific properties of ice.

In view of in-ice navigation of melting probes as described in Kowalski et al. (2016), our results confirm the possibility of the transmission of acoustic signals over tens of meters and thus allowing the determination of the position of a melting probe by

the trilateration of acoustic signals. From our observations lower frequencies below 20 kHz or even below 5 kHz are preferable for this application.

For the application of sub-glacial exploration, e.g. of deep sub-glacial lakes in Antarctica or a space mission to the moon Enceladus, the here observed attenuation would not allow for a navigation volume with sides much larger than typically 100 m. However, the ice quality in other environments can be much improved. E.g. Abbasi et al. (2011) observe an attenuation of about

300 m in deep Antarctic ice. This would allow for a much larger propagation distance of sound and consequently a much larger navigation volume that scales with the cube of the maximum propagation distance. The feasibility of acoustic trilateration for the navigation in the ice shield of Enceladus remains an open question that depends strongly on the modeling of the local glacial environment. An ice-structure deviating from that of alpine glaciers could strongly enhance the performance of such a navigation system.

Concluding, the here presented measurement of the acoustic attenuation length is robust in terms of systematic uncertainties. The obtained values are encouraging for the development and the use of sonographic technologies for the exploration of natural glaciers, even in the presence of cracks and crevasses. An improved theoretical understanding of the effective damping of sound during propagation in such natural glaciers would allow determining whether the measured attenuation and its frequency dependency can be beneficial in characterizing basic properties of the glacier and its ice.



*Code and data availability.* The *in-situ* data are stored in the format of `ROOT` trees and is pre-processed with tools from the ROOT framework (Brun et al., 2018). The final analysis is done by a series of custom scripts in the `python` (Python Software Foundation, 2018) programming language using tools from the publicily available library `NumPy` (NumPy Developers, 2018). The analysis itself is is documented in more details in (Meyer, 2018). Data and example scripts can be obtained on request from the authors.

5   *Author contributions.* The experimental setup has been designed by all signing authors who have contributed either to the preparation of the setup or the measurements on the glacier or both. The data-analysis has been conducted by AM. The methods and results have been reviewed and approved by all authors. The manuscript has been prepared by AM and CW and has been reviewed and approved by all authors.

*Competing interests.* The authors declare that they have no conflict of interest.

*Acknowledgements.* We would like to thank Markus Bobbe (TU Braunschweig) for providing the photograph from Fig. 2. This work has
10   been accomplished within the framework of the Enceladus Explorer Initiative that is managed by the DLR Space Administration. The EnEx-RANGE project is funded by the German Federal Ministry of Economics and Energy (BMWi) by resolution of the German Federal Parliament under the funding code 50NA1501.



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
