# Peer review of "Attenuation of Sound in Glacier Ice from 2 kHz to 35 kHz"

_The Cryosphere, 2018_

## Referee Comment (RC1) · Loewe (Referee) · 17 Dec 2018

**General comments**

The paper investigates the attenuation coefficient of sound waves in ice conducted on a glacier. Overall, the paper presents a very careful description and analysis of an acoustic wave propagation experiment carried out on the Langenferner glacier. I am not an expert in seismic measurements, so I cannot entirely judge the claimed improvement of these experiments over previous work. But the measurement protocol appears to be sound and the study is without a doubt a very careful piece of work so I recommend publication after some revisions have been made.

As a main weakness of the work, no additional, constraining data (texture, porosity, temperature etc) from the ice at the measurement locations was collected which renders the interpretation of the results a bit difficult. Accordingly, the discussion in view of existing work and potential mechanisms remains a bit fuzzy to me and requires a polish. The respective questions are included in the list of specific comments below.

Henning Löwe

**Specific comments**

(p.1 l.6): here presented results → results presented here

(p.2 l.9): polycrystaline → polycrystalline

(p4. l.21): maybe I missed it but *when* was the field campaign carried out?

(p.10 l.19): N → $N$

(p.11 l.27): what does sup stand for?

(Fig 6/7): should be combined to a single figure

(p.15 l.4): a reference should be given for the used method

(p.20 l.11): the wave lengths ($\approx 9 - 60$ cm) as estimated from frequencies and measured speed of sound should be stated somewhere explicitly (not necessarily here, but the occurrence of "wave length" reminds me of that) I think its helpful for the discussion later.

(p.20 l.22): The statement about the comparison to Westphal in the frequency dependence is not clear. From which part of Fig 15 does this follow?

(p.20 l.24): I cannot follow why the present data is not consistent with Rayleigh scattering. Here it seems necessary to recall the prediction of Rayleigh scattering on the frequency dependence and maybe include an inset in Fig 15 to show how this compares to the collected data. In addition, the discussion and comparison to other work should be a bit more comprehensive in view of the similarities in view of of temperature, depth, ice porosity, etc. Given the range of wave lengths, the origin of attenuation by dissipative or scattering mechanisms may be quite different.

(p.20 l.29): Again, the conclusion about the frequency dependence is appears to be an overstatement if numbers (or figures) are not shown.

(p.20 l.32): accounts → account

(p.20 l.32): which differences?

(p.21 l.2): Isn't it possible to discuss/include at least the prediction of the attenuation coefficient/length (maybe derived from the "quality factor" as often used in the geo context) for homogeneous, polycrystalline ice in Fig 15?

(p.21 l.17): Acoustic scattering in heterogeneous materials is reasonably well understood, but it needs additional measurements to characterize the heterogeneities and the state of the material to infer potential origins.

---

## Referee Comment (RC2) · Anonymous Referee #2 · 19 Dec 2018

Review of "Attenuation of Sound in Glacier Ice from 2 kHz to 35 kHz"

This manuscript presents a novel experimental design to measure attenuation in glacier ice in situ. The authors use ultrasonic transducers to collect high-frequency waveforms (chirps, etc.) at varying distances across a glacier. The authors design the survey in such a way that they can characterize the errors in the measurements due to different components of the system (e.g. structural heterogeneity of the glacier ice). The manuscript covers the acquisition system and survey design in detail. A comprehensive error analysis is presented and errors are propagated through to the final estimate of ice attenuation amplitudes. However, I find the conclusions rather lacking, especially the comment regarding the attenuation mechanism as it relates to Rayleigh scattering. I think the authors would do well to reconsider this conclusion and really make an effort

to discuss their reasoning and evidence for this conclusion. I have included my main remarks below, as well as included an annotated PDF so that the authors can improve the English grammar and writing style.

Pg 3 L14 (point 6): The water is necessary to propagate the compression wave. It is also there to keep the hole open I would assume and occurs no matter what because of the drilling method. I do not see the need for this statement. Why not just say water is present in the hole outside of this enumerated list? For instance on page 5 line 5 can be used for this.

Figure 1: It would be great to have a map inset to see what in Italy this is located.

Page 4 L15: What was the surface-air temperature during these experiments?

Paragraph structure (for example the first sentence in Section 2.4): A single sentence is not a paragraph. Please revise these sentences throughout the manuscript.

Why is the electronic noise so strong? Did you use shielded cables? Was the excess cable wrapped in loops?

Pg. 9 L5: Is the crosstalk in the source signal as well? If it is, then how can you remove that cross talk from the amplitudes before you normalize?

Pg 9 L26: What does the following sentence actually mean? It does not make sense to me. "The noise estimate in the noise window matches the noise-level for the signal window reasonably well."

Throughout document: Please use emitter and do not switch between emitter and "sender." This is confusing. You do the same thing with sensor and receiver. Please stick to receiver.

Pg 10 last line: Where is the normalization by N to make this equation represent a mean? Also, the \sigma_i^2 terms cancel, so how is this an error weighted mean?

Equation 1: Why is there not a subscript i (i.e. \sigma_i) on the left-hand side of the

equation? I think there should also be m and n subscripts for this error estimate as well.

In the data processing you do not mean revamping the mean? It is possible that a DC component to the data accumulates over time and that leads to the variation you see in Figure 7, rather than spontaneous changes in the ice? You mention windowing, but not linear or mean detrending. These are common steps in waveform data processing. It would also be interesting to know the air temperature during this time. The drop off in amplitude in Figure 7 at 18h is quite dramatic.

Is the time in plots local time or GMT time? It would be most useful if they were in local time.

Pg 14 L2: The given distances of 10m, 60m, and 90m do not figure 10. Please fix.

Pg 16 L18: Is this variation due to fabric-induced anisotropy? If so, can you please discuss. The term "glacier geomorphology" is not very intuitive as it pertains to sound speed. I do not think readers will understand how geomorphology can cause velocity variations. I am not sure that I understand what you mean here.

You discuss the influence of temperature changes on your measurements, but you do not cite recent and relevant work that studied attenuation as a function of temperature: "Monitoring the temperature-dependent elastic and anelastic properties in isotropic polycrystalline ice using resonant ultrasound spectroscopy", https://www.the-cryosphere.net/10/2821/2016/tc-10-2821-2016.html

Your final comment on Rayleigh scattering in the conclusion section seems unfounded. You reference the Westphal 1965 paper in your introduction, do some experiments, and then say, "look, we found it is not Rayleigh scattering". This is not rigorous, nor is it convincing. You pose no other mechanism and it seems like you would do the community a favor by providing a discussion as to why you think Rayleigh scattering is not the mechanism. Even explaining to the reader what Rayleigh scatter is would be a

useful first step. Are you making this claim simply because your data do not follow an attenuation of frequency to the 4th power?

Please also note the supplement to this comment:
https://www.the-cryosphere-discuss.net/tc-2018-224/tc-2018-224-RC2-supplement.pdf

**Supplement:**

[Figure]

[Figure]

**Attenuation of Sound in Glacier Ice from 2 kHz to 35 kHz**

[revised manuscript text omitted]

---

## Author Comment (AC1) · 2 Jan 2019

Dear collegue,

we thank you for your in-depth review and many helpful comments. Based on your and the other reviewers comments we have substantially revised and reformulated the paper in many place and fixed language issues. We hope that we have addressed appropriately your feedback.

For you information we have uploaded our response to both reviews as acrobat comments included in your review and have also included the current paper-draft where we have implemented all changes.

Best regards, Christopher Wiebusch for the authors

[Figure]

Please also note the supplement to this comment:
https://www.the-cryosphere-discuss.net/tc-2018-224/tc-2018-224-AC1-supplement.zip

―――――――――――――――――

---

## Author Response (AR1)

**General comments**

The paper investigates the attenuation coefficient of sound waves in ice conducted on a glacier. Overall, the paper presents a very careful description and analysis of an acoustic wave propagation experiment carried out on the Langenferner glacier. I am not an expert in seismic measurements, so I cannot entirely judge the claimed improvement of these experiments over previous work. But the measurement protocol appears to be sound and the study is without a doubt a very careful piece of work so I recommend publication after some revisions have been made.

As a main weakness of the work, no additional, constraining data (texture, porosity, temperature etc) from the ice at the measurement locations was collected which renders the interpretation of the results a bit difficult. Accordingly, the discussion in view of existing work and potential mechanisms remains a bit fuzzy to me and requires a polish. The respective questions are included in the list of specific comments below.

Henning Löwe

**Specific comments**

(p.1 l.6): here presented results → results presented here

(p.2 l.9): polycrystaline → polycrystalline

(p4. l.21): maybe I missed it but *when* was the field campaign carried out?

(p.10 l.9): N → $N$

(p.11 l.27): what do s sup stand for?

(Fig 6/7): should be combined to a single figure

(p.15 l.4): a reference should be given for the used method

(p.20 l.11): the wave lengths ($\approx 9 - 60$ cm) as estimated from frequencies and measured speed of sound should be stated somewhere explicitly (not necessarily here, but the occurrence of "wave length" reminds me of that) think its helpful for the discussion later.

(p.20 l.22): The statement about the comparison to Westphal in the frequency dependence is not clear. From which part of Fig 15 does this follow?

(p.20 l.24): I cannot follow why the present data is not consistent with Rayleigh scattering. Here it seems necessary to recall the prediction of Rayleigh scattering on the frequency dependence and maybe include an inset in Fig 15 to show how this compares to the collected data. In addition, the discussion and comparison to other work should be a bit more comprehensive in view of the similarities in view of of temperature, depth, ice porosity, etc. Given the range of wave lengths, the origin of attenuation by dissipative or scattering mechanisms may be quite different.

(p.20 l.29): Again, the conclusion about the frequency dependence is appears to be an overstatement if numbers (or figures) are not shown.

(p.20 l.32): counts → account

(p.20 l.32): which differences?

(p.21 l.2): Is it possible to discuss/include at least the prediction of the attenuation coefficient/length (maybe derived from the "quality factor" as often used in the geo context) for homogeneous, polycrystalline ice in Fig 15?

**Number: 1**    Author: wiebusch    Subject: Hervorheben    Date: 28.12.2018 17:13:24
Yes we agree that this is a weak point.
Our focus was a robust measurement that was hard to obtain. Additional data from the same glacier may become avalibale in the future by our collegues J. Kowalski who has measured temperature profiles and S.Galos et.al. who continously works on this specific glacier.

**Number: 2**    Author: wiebusch    Subject: Hervorheben    Date: 02.01.2019 14:49:47
We did polish the discussion.

**Number: 3**    Author: wiebusch    Subject: Hervorheben    Date: 28.12.2018 12:24:03
fixed,, removed here

**Number: 4**    Author: wiebusch    Subject: Hervorheben    Date: 28.12.2018 12:25:17
fixed, added also hyphen

**Number: 5**    Author: wiebusch    Subject: Hervorheben    Date: 28.12.2018 12:31:35
added August 2017

**Number: 6**    Author: wiebusch    Subject: Hervorheben    Date: 28.12.2018 12:33:15
fixed

**Number: 7**    Author: wiebusch    Subject: Hervorheben    Date: 28.12.2018 17:14:08
Wiki: The supremum (abbreviated sup; plural suprema) of a subset S of a partially ordered set T is the least element in T that is greater than or equal to all elements of S, if such an element exists.Consequently, the supremum is also referred to as the least upper bound (or LUB).

so the value is 0 unless the second argument is >0, then it is the second.

**Number: 8**    Author: wiebusch    Subject: Hervorheben    Date: 28.12.2018 12:40:03
difficult, because this is distinct data that cannot be easily compared. The measurement time and span and measured amplitudes differ.
We could put the figures side-by-side.

We would need a specific suggestion  how to combine.

**Number: 9**    Author: wiebusch    Subject: Hervorheben    Date: 28.12.2018 12:49:04
fixed

**Number: 10**    Author: wiebusch    Subject: Hervorheben    Date: 28.12.2018 13:28:48
we added right at the beginning that 1-100kHz coprresponds to 350-3.5 cm

**Number: 11**    Author: wiebusch    Subject: Hervorheben    Date: 28.12.2018 17:15:28
westphal is not included in the figure. text is now modified to make this clear

**Number: 12**    Author: wiebusch    Subject: Hervorheben    Date: 28.12.2018 17:16:19
for rayleigh scattering we would expect an attenuation length dependence with the fourth power.
Our result is more in agreement with internal friction as suggested in the literatureto be the dominating effect in warm ice.

**Number: 13**    Author: wiebusch    Subject: Hervorheben    Date: 28.12.2018 14:34:49
we have tried to do so in the text

**Number: 14**    Author: wiebusch    Subject: Hervorheben    Date: 28.12.2018 17:16:58
it seems that  our measurement is quite consistent with dissipative loss. That thenm should be rather similar for different "warm" ice.

**Number: 15**    Author: wiebusch    Subject: Hervorheben    Date: 28.12.2018 14:37:34
improved the text

**Number: 16**    Author: wiebusch    Subject: Hervorheben    Date: 28.12.2018 14:26:43
fixed

**Number: 17**    Author: wiebusch    Subject: Hervorheben    Date: 28.12.2018 17:17:18
the attenuation is a factor 10 as discussed in the introduction

**Number: 18**    Author: wiebusch    Subject: Hervorheben    Date: 28.12.2018 17:17:40
as discussed in the introduction theoretical predictions have not been very succesful: being wrong by orders of magnitude for simple polycristaline ice.

(p.21 l.17): 1coustic scattering in heterogeneous materials is reasonably well understood, but it needs additional measurements to characterize the heterogeneities and the state of the material to infer potential origins.

Number: 1     Author: wiebusch     Subject: Hervorheben     Date: 28.12.2018 14:23:02

Yes, we agree. But it is not necessarily scattering.

Changed to:

An improved  understanding of the effective damping of sound in natural glaciers is required before  the attenuation and its frequency dependence can be beneficial in characterizing basic properties of the  glacier ice.

This will require to combine attenuation measurements with measurements of glacial parameters that characterize the heterogeneity and also to study temperature-dependent effects.

The Cryosphere Discuss.,
https://doi.org/10.5194/tc-2018-224-RC2, 2018
This manuscript presents a novel experimental design to measure attenuation in glacier ice in situ. The authors use ultrasonic transducers to collect high-frequency waveforms (chirps, etc.) at varying distances across a glacier. The authors design the survey in such a way that they can characterize the errors in the measurements due to different components of the system (e.g. structural heterogeneity of the glacier ice). The manuscript covers the acquisition system and survey design in detail. A comprehensive error analysis is presented and errors are propagated through to the final estimate of ice attenuation amplitudes. However,
[Figure]
 I find the conclusions rather lacking, especially the comment regarding the attenuation mechanism as it relates to Rayleigh scattering. I think the authors would do well to reconsider this conclusion and really make an effort

to discuss their reasoning and evidence for this conclusion. I have included my main remarks below, as well as included an annotated PDF so that the authors can improve the English grammar and writing style.

[2] Pg 3 L14 (point 6): The water is necessary to propagate the compression wave. It is also there to keep the hole open I would assume and occurs no matter what because of the drilling method. I do not see the need for this statement. Why not just say water is present in the hole outside of this enumerated list? For instance on page 5 line 5 can be used for this.

Figure 1: It would be great to have a map inset to see what in Italy this is located [3]

Page 4 L15: What was the surface-air temperature during these experiments? [4]

Paragraph structure (for example the first sentence in Section 2.4): A single sentence is not a paragraph. Please revise these sentences throughout the manuscript. [5]

Why is the electronic noise so strong? Did you use shielded cables? Was the excess cable wrapped in loops? [6]

Pg. 9 L5: Is the crosstalk in the source signal as well? If it is, then how can you remove that cross talk from the amplitudes before you normalize? [7]

Pg 9 L26: What does the following sentence actually mean? It does not make sense to me. "[8] The noise estimate in the noise window matches the noise-level for the signal window reasonably well."

[9] Throughout document: Please use emitter and do not switch between emitter and "sender." This is confusing. You do the same thing with sensor and receiver. Please stick to receiver.

Pg 10 last line: Where is the normalization by N to make this equation represent a mean? Also, the $\sigma_i^2$ terms cancel, so how is this an error weighted mean? [10]

Equation 1: Why is there not a subscript i (i.e. $\sigma_i$) on the left-hand side of the

**Page: 4**

**Number: 1**     Author: wiebusch    Subject: Hervorheben     Date: 28.12.2018 17:18:58

we have substantially reworked  the discussion

**Number: 2**     Author: wiebusch    Subject: Kommentar zu Text   Date: 27.12.2018 17:44:31

Added:
The water interface is advantageous compared to dry holes because it
improves the coupling of the transducers to the ice.

**Number: 3**     Author: wiebusch    Subject: Notiz     Date: 27.12.2018 18:44:48

Here you are:
https://www.google.com/maps/place/Rifugio+Casati+al+Cevedale+mt+3269/@46.4703661,10.5718486,13z/data=!4m5!3m4!
1s0x0:0xc0afb8a88f5d1295!8m2!3d46.463158!4d10.602489

We prefer not to change the figure to an even smaller scale., and the gegarphic locations are well defined.

**Number: 4**     Author: wiebusch    Subject: Notiz     Date: 27.12.2018 18:43:30

outside air was up to +10°C during day but below 0° during night.

**Number: 5**     Author: wiebusch    Subject: Notiz     Date: 27.12.2018 18:39:16

OK

**Number: 6**     Author: wiebusch    Subject: Notiz     Date: 27.12.2018 18:54:13

Sure, cables are shielded and not looped, except for a few simple connectors. However, we generate 500V pulses for the largest distances inside
the same DAQ box to which the signal comes back.
We think, that the observed cross-talk on the few 10 mV level is actually quite good.

**Number: 7**     Author: wiebusch    Subject: Notiz     Date: 27.12.2018 18:50:45

The cross talk is generated by the source signal. We do measure the amplitude of this signal and normalize the received acoustic signal to the
emitted amplitude.
This normalization is not affected by cross talk.

**Number: 8**     Author: wiebusch    Subject: Hervorheben     Date: 27.12.2018 19:04:02

Changed to:
The noise-level estimated from the noise window matches the apparent noise-level from the signal window reasonably well.

**Number: 9**     Author: wiebusch    Subject: Hervorheben     Date: 27.12.2018 19:37:58

fixed, thanks

**Number: 10**     Author: wiebusch    Subject: Notiz     Date: 28.12.2018 17:21:00

These are standard text book formulas.
The sigma's are inside the sum and do not cancel. If all sigma are the same, you get an division by N.

equation? I think there should also be m and n subscripts for this error estimate as well.
[Figure]

In the data processing you do not mean revamping the mean?
[Figure]
It is possible that a DC component to the data accumulates over time and that leads to the variation you see in Figure 7, rather than spontaneous changes in the ice? You mention
windowing, but not linear or mean detrending. These are common steps in waveform data processing. It would also be interesting to
know the air temperature during this time. The drop off in amplitude in Figure 7 at 18h is quite dramatic.

Is the time in plots 5 local time or GMT time? It would be most useful if they were in local time.

Pg 14 L2: The given distances of 10m, 60m, and 90m do not figure 10. Please fix. 6

Pg 16 L18: Is this variation due to fabric-induced anisotropy? If so, can you please discuss. The term "glacier geomorphology" is not very intuitive as it pertains to sound speed. I do not think readers will understand how geomorphology can cause velocity variations. I am not sure that I understand what you mean here.

You discuss the influence of 7 temperature changes on your measurements, but you do not cite recent and relevant work that studied attenuation as a function of temperature: "Monitoring the temperature-dependent elastic and anelastic properties in 8 isotropic polycrystalline ice using resonant ultrasound spectroscopy", https://www.the-cryosphere.net/10/2821/2016/tc-10-2821-2016.html

9 Your final comment on Rayleigh scattering in the conclusion section seems unfounded. You reference the Westphal 1965 paper in your introduction, do some experiments, and then say, "look, we found it is not Rayleigh scattering". This is not rigorous, nor is it convincing. You pose no other mechanism and it seems like you would do the community a favor by providing a discussion as to why you think Rayleigh scattering is not the mechanism. Even explaining to the reader what Rayleigh scatter is would be a

useful first step. Are you making this claim simply because your data do not follow an 10 attenuation of frequency to the 4th power?

Please also note the supplement to this comment:
https://www.the-cryosphere-discuss.net/tc-2018-224/tc-2018-224-RC2- 11
supplement.pdf

**Page: 5**

**Number: 1**      Author: wiebusch     Subject: Notiz      Date: 27.12.2018 19:55:15

one could write a subscript i, but we have avoided to do so, to make sure it is the total error for a given measurement point.

n and m subscripts are not needed

**Number: 2**      Author: wiebusch     Subject: Hervorheben      Date: 28.12.2018 17:24:28

We do not accumulate data but rather combine repeated measurements.
Since we work in Fourier space, it is unclear how DC offsets could accumulate over time.
Inspecting the measured raw wave-forms, we have not seen any indication of a DC shift.
Looking at all the raw data, we have not found any convincing artifact in the measured data but basically the amplitudes do change. The time is in local time but effects show no evident correlation with daytime - i.e. human activity. Only sub-dominant noise rates are slightly higher and transients more frequent during daytime.

Your point in terms of environmental temperature is very interesting. During daytime melting water flows over the glacier and during night the surface refreezes. We see no systematic effect between different measurements but, each hole may be differently affected by the day-night breathing of the glacier. Multiple day measurement of a fixed hole could give more insights. Sorry we did not take that data.

So, in the absence of any indication of a measurement issue, we have to interpret this effect as a property of the propagated signal .

The situation is unsatisfying, but we do account for this systematics in the error budget.

**Number: 3**      Author: wiebusch     Subject: Hervorheben      Date: 28.12.2018 17:24:43

windows are chosen very robust safely within the region of interest.
small changes in the propagation speed could not explain such a strong effect

**Number: 4**      Author: wiebusch     Subject: Hervorheben      Date: 27.12.2018 20:12:14

ir temperatures are not recorded but are not inphase with the observed changes. Sun-set is later than 18:00

**Number: 5**      Author: wiebusch     Subject: Hervorheben      Date: 28.12.2018 17:25:01

it's local

**Number: 6**      Author: wiebusch     Subject: Notiz      Date: 28.12.2018 10:03:30

this comments is unclear. the noted distances are included in figure 10

**Number: 7**      Author: wiebusch     Subject: Hervorheben      Date: 28.12.2018 17:25:59

no we did add in the discussion Vaughan et al as you suggest below

**Number: 8**      Author: wiebusch     Subject: Notiz      Date: 28.12.2018 10:07:36

thank you for pointing us to that reference.

**Number: 9**      Author: wiebusch     Subject: Hervorheben      Date: 28.12.2018 17:26:23

that part is substantially reworked

**Number: 10**      Author: wiebusch     Subject: Notiz      Date: 28.12.2018 17:27:33

yes, that would be expected (see price et al.) . maybe not a strict power of 4, to account for additional effects but a strong dependence as is the claim in westphal is not observed here. We have reworked the discussion.

**Number: 11**      Author: wiebusch     Subject: Notiz      Date: 28.12.2018 17:27:56

Thank you for the very detailed review. We comment iyour comments there

[Figure]

**Attenuation of Sound in Glacier Ice from 2 kHz to 35 kHz**

Alexander Meyer[1], Dmitry Eliseev[1], Dirk Heinen[1], Peter Linder[1], Franziska Scholz[1], Lars Steffen Weinstock[1], Christopher Wiebusch[1], and Simon Zierke[1]

[1]III. Physikalisches Institut B, RWTH Aachen University, Otto Blumenthal Str., 52074 Aachen, Germany

**Correspondence:** Christopher Wiebusch (wiebusch@physik.rwth-aachen.de)

**Abstract.** The acoustic damping of sound waves in natural glaciers is a largely unexplored physical property that has relevance for various applications. We present measurements of the attenuation of sound in ice with a dedicated measurement setup *in situ* on the Italian glacier *Langenferner*. The tested frequency ranges from 2 kHz to 35 kHz and probed distances between 5 meter and 90 meter. The attenuation length has been determined by two different methods and detailed investigations of systematic uncertainties. The attenuation length decreases slowly with increasing frequencies. Observed values range between 13 meter for low frequencies and 5 meter for high frequencies. The here presented results strongly improve in accuracy with respect to previous measurements. However, quantitatively the found attenuation is remarkably similar to observations at very different locations.

**1 Introduction**

The acoustic properties of ice are of interest for a large variety of applications ranging from the measurement of seismic waves (Robinson, 1968) to the detection of ultra-high-energy neutrinos (Abbasi et al., 2010). Recently, the application of sonographic methods has received increased interest in the context of the exploration of subglacial lakes in Antarctica or even water oceans below the ice surfaces of moons in the outer solar system. Particularly the joint research collaboration *Enceladus Explorer*, (Kowalski et al., 2016) has developed a maneuverable melting probe in glacial ice. It incorporates two acoustic systems operating in the range of 1 kHz to 1000 kHz. One is based on trilateration of the arrival times of acoustic signals from pingers and allows for the localization of the probe. The other system is based on phased piezo arrays and is used for the sonographic fore-field reconnaissance e.g. the detection of obstacles on the planned trajectory or water pockets when approaching the region of interest.

In water, sonographic imaging and acoustic localization techniques are well established technologies. In ice, however, acoustic navigation techniques are largely unexplored though they may provide a number of applications. Unlike water, not only pressure waves but also shear waves can propagate in the solid state ice. Since pressure waves are easier to generate and have a faster propagation speed (Vogt et al., 2008; Abbasi et al., 2010), they seem more suited for navigation purposes and are focused on in the following.

A limiting parameter is the damping of acoustic signals with distance, that strongly depends on the respective glacial environment and the frequency of the signal. In the following we refer to the attenuation length as that distance $r$ at which the amplitude of a spherical signal is reduced by $1/e$ after correcting the amplitude for the $1/r$ reduction. This parameter itself is

**Number: 1** Author: anonymous Subject: Highlight Date: 15.12.2018 19:44:48

not a complete sentence and does not make sense.

> Author: wiebusch Subject: Notiz Date: 27.12.2018 17:25:38
> fixed: including

**Number: 2** Author: anonymous Subject: Highlight Date: 15.12.2018 19:45:21

What does "slowly" mean? Why not remove this term?

> Author: wiebusch Subject: Notiz Date: 27.12.2018 17:26:49
> fixed

**Number: 3** Author: anonymous Subject: Inserted Text Date: 15.12.2018 19:45:36

s

**Number: 4** Author: anonymous Subject: Cross-Out Date: 15.12.2018 19:45:43

**Number: 5** Author: anonymous Subject: Cross-Out Date: 15.12.2018 19:45:50

**Number: 6** Author: anonymous Subject: Inserted Text Date: 15.12.2018 19:45:37

s

**Number: 7** Author: anonymous Subject: Highlight Date: 15.12.2018 19:46:34

Poor grammar in this sentence. Consider revising.

> Author: wiebusch Subject: Notiz Date: 27.12.2018 17:29:53
> However, the observed attenuation is found remarkably similar to observations at very different locations.

**Number: 8** Author: anonymous Subject: Cross-Out Date: 15.12.2018 19:47:49

**Number: 9** Author: anonymous Subject: Inserted Text Date: 15.12.2018 19:50:23

which

**Number: 10** Author: anonymous Subject: Inserted Text Date: 27.12.2018 17:32:05

due to geometric spreading

an interesting physical property as it depends on both the structures on scales of the wave-length and smaller but effectively integrated over the overall glacial structure. For the purpose of navigation it ultimately limits the maximum distance to which pairs of receiver and emitters can exchange signals. The design and optimization of acoustic transducers of high emission power strongly depends on the frequency and prefers higher frequencies as well as a better beam resolution of phased arrays

5   does.

The acoustic attenuation length in ice is not well known in the range from $1\,\text{kHz}$ to $100\,\text{kHz}$. While in water the attenuation length in this frequency range exceeds orders of kilometers (Fisher and Simmons, 1977; Schulkin and Marsh, 1962) and only slightly varies with temperature and chemical composition, the attenuation in the solid state material ice is more complicated. Even for simple polycrystaline ice, calculations range over orders of magnitude from a few 10 meters to several kilometers

10  depending on the temperature and assumed grain sizes (Price, 2006, 1993).

In a natural glacier environment the situation is even more complicated. Ice cracks filled with air and inclusions of dust and rocks will attenuate sound strongly. Their occurrence depends the general environmental conditions of the glacier such as its formation and flow.

Only few *in situ* measurements exist in the literature for very different glacial environments. The largest measured attenuation

15  length is consistent with about $300\,\text{m} \pm 20\,\%$. It has been observed for the glacial ice at depths $190\,\text{m}$ to $500\,\text{m}$ below the surface at the geographical South Pole, for frequencies between $10\,\text{kHz}$ to $30\,\text{kHz}$. This attenuation is however substantially stronger than the earlier predictions (Price, 2006). Measurements in sea ice by Langleben (1969) for $10\,\text{kHz}$ to $500\,\text{kHz}$ resulted in the range of $9\,\text{m}$ to $2\,\text{m}$ for $10\,\text{kHz}$ to $30\,\text{kHz}$. For frequencies $>100\,\text{kHz}$ see also Lebedev and Sukhorukov (2001). Mesurements of seismic explosion shocks in a temperate glacier are reported in Westphal (1965). These measurememts result

20  in an amplitude attenuation length that ranges between $70\,\text{m}$ to $4.6\,\text{m}$ for frequencies from $2.5\,\text{kHz}$ to $15\,\text{kHz}$. This strong frequency dependency is interpreted as Rayleigh scattering on ice grains as dominant attenuation process. Recent measurements on the alpine glaciers *Morteratsch* and *Pers* (Helbing et al., 2016; Kowalski et al., 2016) with acoustic transducers reported an attenuation of similar scale with a length of $31\,\text{m}$ for $5\,\text{kHz}$ and $15\,\text{m}$ for $18\,\text{kHz}$. Goal of this work has been provide a robust measurement that properly addresses and reduces experimental uncertainties with respect to previous measurements.

25  The measurement of the sound attenuation of sound *in situ* is in fact challenging the accuracy is limited by the quality of the measurement setup and the systematic uncertainties related to the environment. In particular two aspects are important. First, sensors and emitters are inserted into the glacier by holes. The structure of such holes depends on the production process. It differs from hole to hole and changes with time, e.g. because the water-level can change with time due to leakage and refreezing of the walls. As result, the acoustic coupling to the ice differs not only from hole to hole but also for repeated

30  measurements in the same holes. Secondly, the natural glacial environment contains cracks and other absorbing structures. The subsurface ice-structure is unknown. The phase of reflected signals e.g. from the surface, depends on the specific emitter-receiver measurement geometry and thus can interfere with the direct acoustic signal.

The basic concept of the here presented measurement addresses these issues. It is based on the deployment of an acoustic emitter and a receiver a few meter deep into the glacier using holes that are produced with a melting probe. From the relative

35  amplitude of the signal registered for different distances we can infer the attenuation length.
* * *
**Number: 1**      Author: anonymous Subject: Highlight    Date: 15.12.2018 20:00:04

again a poorly constructed sentence that make the meaning hard to understand. please revise.

> **Author: wiebusch**    Subject: Notiz      Date: 27.12.2018 17:36:34
> This parameter itself is an interesting physical property as it depends on small structures on scales of the wave-length but at the same time effectively integrates the overall glacial structure.
* * *
**Number: 2**      Author: anonymous Subject: Highlight    Date: 15.12.2018 20:02:26

revise

> **Author: wiebusch**    Subject: Notiz      Date: 27.12.2018 17:56:37
> will strongly attenuate sound. This will depend on
> the overall environmental conditions
> of the glacier such as its formation and flow.
* * *
**Number: 3**      Author: anonymous Subject: Highlight    Date: 19.12.2018 23:39:56

citation?

> **Author: wiebusch**    Subject: Notiz      Date: 27.12.2018 18:01:59
> fixed, abbasi et al 2011
* * *
**Number: 4**      Author: anonymous Subject: Cross-Out   Date: 15.12.2018 20:03:08
* * *
**Number: 5**      Author: anonymous Subject: Inserted Text       Date: 15.12.2018 20:03:56

, respectively
* * *
**Number: 6**      Author: anonymous Subject: Inserted Text       Date: 27.12.2018 18:04:01

, respectively
* * *
**Number: 7**      Author: anonymous Subject: Inserted Text       Date: 15.12.2018 20:06:31

is
* * *
**Number: 8**      Author: anonymous Subject: Inserted Text       Date: 15.12.2018 20:06:19

The
* * *
**Number: 9**      Author: anonymous Subject: Cross-Out   Date: 15.12.2018 20:06:55
* * *
**Number: 10**      Author: anonymous Subject: Inserted Text       Date: 15.12.2018 20:07:06

,
* * *
**Number: 11**      Author: anonymous Subject: Cross-Out   Date: 15.12.2018 20:08:11
* * *
[Figure]

In order to produce an as robust result as possible, we have established the following strategy:

1.  In all measurement the same pair of sender and receiver is used. Therefore the emitter and receiver sensitivities cancel in the ratio of received signals of different distances.

2.  We use an emitter and a receiver that are largely spherically symmetric in emissivity ($<1\,\mathrm{dB}$ at $18\,\mathrm{kHz}$ according to the manufacturer) and also in sensitivity. This reduces systematic differences due to variations of the orientation of the instruments in the holes for different measurements.

3.  We perform our measurements for a large number of distances from $5\,\mathrm{m}$ to $90\,\mathrm{m}$. This allows for the determination of the attenuation with a large lever arm of multiples of the attenuation lengths as well as the suppression of local glacial effects like cracks or reflections.

4.  We include multiple measurements for the same distance but different locations an and depths in the glacier for the estimation of systematic uncertainties related to local properties of the glacier and reflections.

5.  We include repeated measurements using the same holes that have been used a few days earlier, or of changed depth below the surface, to include uncertainties related to changing hole properties and thus acoustic coupling to the ice.

6.  In each measurement, emitters and receivers are covered by a column of melted water at the bottom of the holes. The water interface improves the reproducibility of the coupling of the transducers to the ice.

7.  We have developed a dedicated electronic setup for this measurement and tested it in the laboratory. The setup produces long signals of sine waves that are thus well defined in frequency. An appropriate time window of the registered sine-burst signals rejects transient ring-in phases until the receiver oscillates in phase as well as phases of electro-magnetic interferences.

8.  In order to match the dynamic range for different distances to our setup, the amplitude of the sender can be changed. The emitted acoustic power is monitored in our setup for each measurement and differences are corrected for in the analysis by normalizing to the amplitude of the emitted signal. This approach also corrects for a possible long-term variation of the electronic setup in terms of gain. The validity of this normalization is verified *in situ* by measurements of different amplitude.

9.  We perform the analysis very carefully by estimating and subtracting noise, identifying systematic uncertainties and a robust error propagation using advanced bootstrapping techniques.
* * *
**Number: 1**      Author: anonymous Subject: Cross-Out   Date: 15.12.2018 20:08:52

> **Author: wiebusch**    Subject: Notiz      Date: 27.12.2018 18:08:10
> produce a robust result
* * *
**Number: 2**      Author: anonymous Subject: Cross-Out   Date: 15.12.2018 20:09:16
* * *
**Number: 3**      Author: anonymous Subject: Inserted Text      Date: 15.12.2018 20:09:30

emitter-receiver
* * *
**Number: 4**      Author: anonymous Subject: Cross-Out   Date: 15.12.2018 20:10:28
* * *
**Number: 5**      Author: anonymous Subject: Inserted Text      Date: 15.12.2018 20:10:36

at
* * *
**Number: 6**      Author: anonymous Subject: Highlight   Date: 15.12.2018 20:14:43

the water is also necessary to propagate the compression wave, no? It is also there to keep the hole open I would assume and occurs no matter what because of the drilling method.

I don't quite see the need for this statement. Why not just say water is present in the hole.

> **Author: wiebusch**    Subject: Notiz      Date: 27.12.2018 17:52:11
> Changed, see response to main points

[Figure]

[Figure]

**Figure 1.** Extended thickness map of the Langenferner glacier based on a modified figure in Stocker-Waldhuber (2010). The Casati hut and camp site of the field test are indicated. Coordinates are in UTM coordinates with east on the x-axis and north on the y-axis

**2 The Measurement Setup**

**2.1 The Langenferner Site**

The Langenferner is a high altitude glacier in the Ortler-Alps in Italy, that extends from its highest point at $3370\,\mathrm{m}$ a.s.l. to the lowest point at $2711\,\mathrm{m}$ a.s.l. at the terminus. Galos (2017) reports a covered area of about $1.6\,\mathrm{km}^2$ (in 2013) and an
5  estimated volume of $0.08\,\mathrm{km}^3$ (in 2010).

   The site of the field test was located in the upper part of the glacier at about $3260\,\mathrm{m}$ a.s.l. close to the Rifugio Casati ($46.46\,°\mathrm{N}|10.60\,°\mathrm{E}$), see Fig. 1. The depth of the glacier in the region of the test site was estimated $90\,\mathrm{m}$ to $100\,\mathrm{m}$ in 2010 (Stocker-Waldhuber, 2010). Based on studies of the mass balance by Galos (2017), the site is part of the ablation zone and the depth was reduced by at least $7\,\mathrm{m}$ since 2010. During the field campaign, the glacier was not covered by snow and the ice
10  could be accessed directly.

   The instrumentation was deployed in the glacier by holes prepared with a $12\,\mathrm{cm}$ diameter melting probe that was developed within the EnEx initiative (Heinen et al., 2017). The layout of the holes at the test site is shown in Fig. 2, their coordinates and depths are detailed in Table 1. The figure shows that the test site includes complex ice structures though the main axis has been largely parallel to the largest visible cracks at the surface.
15  Inside the holes we have measured temperatures close to $0\,°\mathrm{C}$ and the glacier appears largely tempered. However, we have observed over night that water surface of holes refroze and in some cases the acoustic transducers froze to the wall of the holes. Therefore domains in the bulk ice of slightly lower temperature cannot be excluded.

**Page: 9**

| | Number: 1 | Author: anonymous | Subject: Inserted Text | Date: 15.12.2018 20:15:40 |
|---|---|---|---|---|

Extent

| | Number: 2 | Author: anonymous | Subject: Inserted Text | Date: 15.12.2018 20:16:51 |
|---|---|---|---|---|

Glacier

| | Number: 3 | Author: anonymous | Subject: Cross-Out | Date: 15.12.2018 20:17:10 |
|---|---|---|---|---|

| | Number: 4 | Author: anonymous | Subject: Inserted Text | Date: 15.12.2018 20:20:00 |
|---|---|---|---|---|

into

| | Number: 5 | Author: anonymous | Subject: Inserted Text | Date: 15.12.2018 20:20:24 |
|---|---|---|---|---|

;

**Table 1.** Measurement holes. Coordinates are given in the UTM coordinate system (notation east|north|up) relative to hole 1 that is located at *(32U:623382.63|5146718.58|3281.84)*.

| # | Pos. [m] | Coordinates [m] | Depth [m] |
|---|---|---|---|
| 1 | 0 | 0.00\|0.00\|0.00 | 2.6 |
| 2 | 5 | -5.02 \| -0.25 \| -0.36, | 1.8 |
| 3 | 10 | -10.09 \| -0.50 \|-0.81 | 2.1 |
| 4 | 30 | -30.27\| -4.20 \| -0.85 | 2.5 , 6[a] |
| 5 | 50 | -50.18 \|-2.75 \| -1.19 | 2.7 |
| 6 | 70 | -70.95 \| -0.91 \| -1.05 | 2.6 |
| 7 | 90 | -90.78 \| 0.47 \| -0.64 | 2.5 |

[a]changed $27^{th}$ August

[Figure]

**Figure 2.** Aerial view of the measurement site with the location of the measurement holes. Modified photo from Markus Bobbe, TU Braunschweig.

**2.2 Instrumentation and setup**

The schematic overview of the measurement setup is shown in Fig. 3. Two spherical, 4.25 inch, acoustic transducers of type ITC-1001 from *International Transducer 1ooperation* are used for sending and receiving the signals. This type of transducer provides a high power broadband acoustic omni-directional emissivity from 2 kHz to 38 kHz and equally good receiving

5   properties. These transducers are connected to the  using coax cables and are lowered into the  water-filled holes. All other  contained in a  metal box on the glacier to shield it from the outdoor environment. In each measurement the  transducer are not inter-changed for emitting and receiving the acoustic signals.

The setup is controlled through Ethernet connections by a notebook running LabVIEW. Signals are generated with a function generator (*Rigol DG5072*), amplified with a power amplifier (*Monacor PA-4040*) and sent to the emitter. The function generator

10   also triggers the data acquisition that is done with a digital oscilloscope (*Tektronix DPO4034*). The signal of the acoustic receiver is amplified and synchronously recorded with this oscilloscope with a sampling rate of 1 MHz. Because of the large

Number: 1    Author: anonymous Subject: Highlight   Date: 15.12.2018 20:25:09
Is this correct? Should it be "corporation"?

Author: wiebusch    Subject: Notiz       Date: 27.12.2018 18:14:05
sorry - automatic spell

Number: 2    Author: anonymous Subject: Cross-Out  Date: 15.12.2018 20:25:20

Number: 3    Author: anonymous Subject: Inserted Text        Date: 15.12.2018 20:25:21
-

Number: 4    Author: anonymous Subject: Cross-Out  Date: 15.12.2018 20:25:58

Number: 5    Author: anonymous Subject: Inserted Text        Date: 27.12.2018 18:15:08
acquisition system

Number: 6    Author: anonymous Subject: Inserted Text        Date: 27.12.2018 18:16:18
components of the acquisition system are

Number: 7    Author: anonymous Subject: Inserted Text        Date: 27.12.2018 18:16:40
weather-proof

Number: 8    Author: anonymous Subject: Cross-Out  Date: 15.12.2018 20:27:42

[Figure]

[Figure]

[Figure]

**Figure 3.** Schematics of the instrument setup.

difference of probed distances the electrical amplitude driving the emitter is dynamically adapted with peak-to-peak amplitudes ranging from $2\,\mathrm{V}$ to $500\,\mathrm{V}$. The LabView program automatically adjusts the dynamic range of the oscilloscope for maximum resolution of the received signal. Furthermore, we measure the power of the emitted signal during each measurement by monitoring the voltage and the current at the emitter input with a $1.1\,\Omega$ power resistor that is connected in series with the

5    emitter. In the data analysis, the normalization of the received acoustic signals is corrected for the different emission power based on these recorded values.

**2.3 Measurement procedures**

Each measurement was carried out according to a strict procedure to ensure consistent data throughout the campaign. The spherical transducers were lowered to the bottom of the holes and were always covered by at least $30\,\mathrm{cm}$ of water. The main

10    attenuation measurement is based on repeated sine bursts of $50\,\mathrm{ms}$ duration. We scan for each pair of holes the frequency band of $2\,\mathrm{kHz}$ to $35\,\mathrm{kHz}$ in steps of $1\,\mathrm{kHz}$. To reduce ambient noise, the repeated burst signals of each frequency are averaged within the oscilloscope as indicated in Table 2. After one full frequency scan, the full procedure is repeated several times.

A measurement window of $100\,\mathrm{ms}$ was selected for the recording of data. This is substantially longer than the signal duration and allows recording $20\,\mathrm{ms}$ of ambient noise before a signal is emitted, and is sufficient to capture the complete signal including

15    a propagation delay of up to $30\,\mathrm{ms}$ that corresponds to a distance of more than $100\,\mathrm{m}$. The burst duration of $50\,\mathrm{ms}$ results in a minimum of 100 oscillations for the lowest frequency. This ensures a sufficiently long stable phase of forced resonance. By appropriate windowing during the offline analysis, phases of unstable amplitudes at the start and end of the burst are omitted. Similarly, phases of electromagnetic interferences are excluded from the analyzed time-windows, as described below.

In addition to these sine bursts, we have regularly recorded *logarithmic chirps* of $3\,\mathrm{ms}$, $5\,\mathrm{ms}$ and $10\,\mathrm{ms}$ duration within

20    frequency ranges between $0.5\,\mathrm{kHz}$ to $42.5\,\mathrm{kHz}$ as well as $11.14$. Barker codes of $10\,\mathrm{kHz}$ and $20\,\mathrm{kHz}$ carrier frequency with

Number: 1           Author: anonymous  Subject: Cross-Out  Date: 15.12.2018 20:31:14

Author: wiebusch    Subject: Notiz        Date: 27.12.2018 18:20:16
the amplitude of the received acoustic signals is corrected for the different emission power based on these recorded values

Number: 2           Author: anonymous  Subject: Inserted Text      Date: 15.12.2018 20:30:45

,

Number: 3           Author: anonymous  Subject: Inserted Text      Date: 15.12.2018 20:31:26

are normalized

Number: 4           Author: anonymous  Subject: Inserted Text      Date: 15.12.2018 20:36:15

-

Author: wiebusch    Subject: Notiz        Date: 27.12.2018 18:22:09
kept this siunitx generated output.

[Figure]

[Figure]

**Table 2.** Measurement runs

| # | Date | Dist. [m] | Holes | Avg. | Rep. | Dur. [hh:mm] |
|---|------|-----------|-------|------|------|--------------|
| 6 | 23.08 | 60 | 6 → 3 | 512 | 3 | 04:33 |
| 7[a] | | 70 | 6 → 1 | 512 | 7 | 11:31 |
| 8[b] | | 10 | 1 → 3 | 512 | 1 | 00:35 |
| 9 | 24.08 | 10 | 1 → 3 | 128 | 4 | 01:53 |
| 10 | | 10 | 3 → 1 | 128 | 4 | 01:58 |
| 11[a] | | 50 | 5 → 1 | 128 | 35 | 17:08 |
| 12 | | 40 | 5 → 3 | 128 | 4 | 01:51 |
| 13 | 25.08 | 20 | 4 → 3 | 128 | 4 | 01:58 |
| 14 | | 30 | 4 → 1 | 128 | 4 | 01:51 |
| 15[c] | | 90 | 7 → 1 | 521 | 2 | 02:09 |
| 16 | | 20 | 5 → 4 | 128 | 4 | 01:52 |
| 17 | 26.08 | 40 | 6 → 4 | 128 | 4 | 01:52 |
| 18[d] | | 60 | 7 → 4 | 128 | 2 | 01:01 |
| 19 [a,c] | | 90 | 7 → 1 | 512 | 13 | 15:33 |
| 20 | | 40 | 7 → 5 | 128 | 4 | 01:52 |
| 21 | 27.08 | 20 | 6 → 5 | 128 | 4 | 01:51 |
| 22 | | 5 | 2 → 1 | 32 | 4 | 00:49 |
| 24[a] | | 60 | 6 → 3 | 512 | 9 | 15:18 |
| 25[e] | | 20 | 4 → 3 | 32 | 6 | 02:02 |
| 26 | 28.08 | 30 | 4 → 1 | 32 | 4 | 01:08 |
| 27 | | 25 | 4 → 2 | 32 | 1 | 00:26 |

[a]During night, [b]100% sending power, sine-bursts 2 kHz to 5 kHz,
25 kHz to 35 kHz only, [c]Sine-bursts 2 kHz to 25 kHz only, [d]Signal
generator switched off, [e]Hole 4 deepened to 6 m

four oscillations per bit (Barker, 1953). These signals are used to determine the speed of sound. The chirps are also used for a second attenuation measurement with independent data.

An overview on the measurement runs that are used for the further data analysis is given in Table 2. Test runs and runs with data failures have been excluded from the list.

[Figure]

**Figure 4.** Waveform from measurement series 12 at 12 kHz (top) and the synchronously measured sending amplitude (bottom). The indicated windows 1 to 3 are relevant for the data analysis and are discussed in the text.

**2.4 Waveform processing and amplitude extraction**

Figure 4 shows as example a recorded waveform from the measurement series 12 for a 12 kHz burst at 40 m distance and the synchronously recorded signal that drives the emitter.

The recorded waveform features several characteristic properties that are explained in the following. From $-20$ ms to $0$ ms pure noise is recorded. Starting with the signal at $0$ ms, we observe a cross-talk from electromagnetic interference in the received signal, that identified due to its lack of propagation delay. After a delay of about $10$ ms the acoustic signal sets in and is interfering with the cross-talk signal. Because the electromagnetic and acoustic signal have a constant relation in relative phase, the superposition is coherent. After $50$ ms the sine-burst is switched off and immediately the interference in the received signal disappears. The now clean acoustic signal continues for the propagation delay up to about $60$ ms, where it stops and the receiver rings down.

**2.4.1 Selection of analysis time windows in the waveforms**

The electro-magnetic interference is caused by the high-power audio amplifier and the sensitive oscilloscope being packed very tightly in the metal box on the glacier. In the field we have verified by unplugging the emission cables that the cross-talk happens locally in the metal box and not at the receiving transducer. The amplitude of the cross-talk has been found to be proportional to the sending amplitude. Note, that the frequency of the electromagnetic and the acoustic signal are the same for each measurement but the relative phase varies due to different propagation delays for different measurement. As result, we have observed both constructive as well as destructive interference between the two signals in the data. For the data analysis
* * *
Number: 1     Author: anonymous Subject: Sticky Note     Date: 15.12.2018 20:38:43

One sentence is not a paragraph.
* * *
Number: 2     Author: anonymous Subject: Cross-Out  Date: 15.12.2018 20:39:57
* * *
Number: 3     Author: anonymous Subject: Inserted Text     Date: 15.12.2018 20:40:22

this
* * *
Number: 4     Author: anonymous Subject: Inserted Text     Date: 15.12.2018 20:40:20

the
* * *
Number: 5     Author: anonymous Subject: Inserted Text     Date: 27.12.2018 18:27:22

arrives (40 m from emitter) with an amplitude above the electronic noise.

Author: wiebusch    Subject: Notiz     Date: 27.12.2018 18:30:34
this change would change the intention of the sentence. Important is the interference with phase correlyted  electronic cross talk of the same frequency.
added elcromagnetic

we therefore use only acoustic data without interference. This can be easily accomplished, because for hole distances $d < 15\,\text{m}$ sending amplitudes are small and received acoustic amplitudes are so large that the cross-talk can be neglected. At larger distances where the sending signal and corresponding cross-talk signal becomes larger, the propagation delay of the acoustic signal allows for a proper separation in time.

5    The selected windows are displayed in the example shown in Fig. 4. For the data processing we have selected for each measurement a window, (2) in Fig. 4, that contains the acoustic signal but no electromagnetic interference. Two windows of the same size are used to determine the noise in the causally unrelated region before the signal, (1) in Fig. 4, and, corrected for the propagation delay, in the recorded sending signal to determine the normalization of the sending signal, (3) in Fig. 4.

For distances $d < 15\,\text{m}$, where the electromagnetic interference is negligible, we chose a signal window which is $20\,\text{ms}$
10  delayed with respect to the start of the acoustic signal (to avoid ring-in effects) and a width of $19\,\text{ms}$. For larger distances, the window starts with a margin of $2\,\text{ms}$ after the end of the $50\,\text{ms}$ long emission burst. The duration of the window depends on the distance assuming a propagation velocity of $3.6\,\text{m}\,\text{ms}^{-1}$ minus a margin of $0.5\,\text{ms}$. For distances of $80\,\text{m}$ and above, the window width is limited to of $19\,\text{ms}$. The proper adjustment of these windows has been applied for each measurement by an automated procedure but has been also visually verified during the analysis.

15  ### 2.4.2 Fourier transformation

In the next step the data in each of the three time windows is Fourier transformed.

Though the three windows are already matched to the same width, they are further optimized with respect to the frequency of the respective sine burst such that exactly $N$ complete periods are inside the window, preventing spectral leakage due to incomplete periods. Furthermore, from the ratio of the signal- and sampling-frequencies the optimum number of data points
20  fitting into this window is estimated. All signal windows are shortened accordingly. The shortening amounts to a maximum $0.5\,\text{ms}$ for the $2\,\text{kHz}$ signal.

Prior to the Fourier transformation, each signal window is multiplied with a Blackman window to further reduce boundary effects and spectral leakage. Since only the amplitude is of interest for the analysis, the absolute values of the Fourier transformation coefficients are taken, discarding the phase information.
25   of the transform is shown
  in Fig. 5 for the largest measured distance of $90\,\text{m}$. The signal clearly exceeds the noise level with a SNR of about 10:1, the noise estimate in the noise window matches the noise-level for the signal window reasonably well. However, a precise prediction based on  different time window cannot be expected because of transient noise fluctuations.

**2.4.3 Noise reduction by spectral subtraction**

30  During the measurements we have observed that the noise level strongly varies with the time of day, i.e. the human activity on the glacier. Therefore the noise is subtracted from the signal Fourier spectrum for each measurement repetition $i$ individually. In order to avoid fluctuations, we average the values of the noise floor in a window $\pm 0.5\,\text{kHz}$ around the respective target

**Page: 14**

Number: 1          Author: anonymousSubject: Cross-Out  Date: 15.12.2018 23:47:01

Number: 2          Author: anonymousSubject: Inserted Text          Date: 15.12.2018 23:47:01
  An example

Number: 3          Author: anonymousSubject: Inserted Text          Date: 15.12.2018 23:52:00
  at a frequency of 9 kHz.

Number: 4          Author: anonymousSubject: Inserted Text          Date: 15.12.2018 23:52:44
  a

[Figure]

[Figure]

**Figure 5.** Frequency spectra for noise and signal windows for a burst measurement during series 19 at $9\,\mathrm{kHz}$.

frequency. The subtraction is performed quadratically $S_i(f) = \sqrt{Y_i^2(f) - \overline{N_i}^2}$, where $Y_i$ is the measured signal and $\overline{N_i}$ is the frequency averaged noise for the repetition i. This is based on the assumption, that the noise is uncorrelated in the time-domain.

We find generally a good SNR for all measurements and the noise subtraction is a rather small correction in most cases. Only for one waveform $Y_i^2(f) < \overline{N_i}^2$ was found, probably due to a strong transient signal overlapping with the measurement.
5  This waveform from measurement series 7 over $70\,\mathrm{m}$ at $29\,\mathrm{kHz}$ has been discarded from the analysis.

Besides the subtraction of noise, the measured noise-level serves as a certainty estimate of the measured signal $S_i$ and and we have used $S_i = \overline{N_i}$.

**2.4.4  Normalization to the emission power**

Synchronously to the measured acoustic data, the sender's voltage $V$ and current $I$ are measured and stored as waveforms as
10  shown in Fig. 4. These waveforms are Fourier transformed as well and the peak sending power $P_i = V \cdot A$ is determined by the multiplied coefficients of the target frequency. The normalized signal amplitude is given by $\hat{S}_i = S_i/\sqrt{\frac{P_i}{2}}$, where the factor $\sqrt{2}$ corrects the peak power to the effective sending power. The uncertainty $\sigma_{S_i}$ is multiplied with the same factor.

In the measurement series 8 and 9 we have verified the correctness of this normalization, by performing the same measurement but changing the emission power by a factor 200 resulting in highly different amplitudes, once close to the detection
15  threshold and once close to saturation. The normalized amplidudes are found fully consistent.

**2.4.5  Data averaging**

The amplitude extraction is repeated for each repetition within one series, see Table 2. We have observed, that particularly during long measurement series both extracted signal and noise level can vary significantly between measurements. Therefore we calculate for each series $n$ the error weighted mean of all N repetitions $S_n = \frac{\sum_{i=1}^{N} S_i/\sigma_i^2}{\sum_{i=1}^{N} 1/\sigma_i^2}$ and the corresponding error $\sigma_n = $

Number: 1       Author: anonymous Subject: Cross-Out   Date: 15.12.2018 23:55:40

Number: 2       Author: anonymous Subject: Inserted Text      Date: 15.12.2018 23:55:32
an

Number: 3       Author: anonymous Subject: Highlight   Date: 15.12.2018 23:56:09
What is this? You do not say. I presume it is the standard deviation, but you should be explicit.

Number: 4       Author: anonymous Subject: Highlight   Date: 15.12.2018 23:58:16
See comment to authors on use of "sender".

[Figure]

[Figure]

$\sqrt{\frac{1}{\sum_{i=1}^{N} 1/\sigma_i^2}}$. Deviations from these averages are assumed to be caused by systematic uncertainties and will be investigated in the following.

**2.5 Stability of data in time**

For the estimation of the total uncertainty of each measurement, we have to take into account several effects

5.  1. Changes of the extracted signal for different repetitions during long measurement series result in an error $\sigma_{S,i}$ of the averaged value in addition to the propagated errors $\sigma_n$.

   2. Differences of the extracted signal for repeated measurements in the same hole but different dates n and m indicate systematic variations of the glacial conditions during the measurement campaign. This additional uncertainty is named $\sigma_{S_{n,m}}$.

10.  3. Differences of the extracted signal ratio for pairs of two holes at the same distance [1]t different positions on the glacier and dates of the measurement [2]dicate the uncertainty related to the local position on the glacier. This additional uncertainty is called $\sigma_{S_n,S_m}$.

The total uncertainty for each signal $S_i$ is then given by

$$\sigma = \sqrt{\sigma_n^2 + \sigma_{S,i}^2 + \sigma_{S_{n,m}}^2 + \sigma_{S_n,S_m}^2},\tag{1}$$

15.  where each uncertainty  [3]lated to the respective effect.

**2.5.1 Observed changes during measurement series**

The repeated measurements during long measurement series allow for the investigation of systematic changes of the measured amplitudes over time. Figure 6 and 7 show [4]  the results from two measurement series of more than $10\,\mathrm{h}$ run time and a large number of repetitions. While the results in the first example are stable within their uncertainties, the second example

20.  shows a systematic variation exceeding the assumed errors.

The origin of this effect remains unclear. However, we can exclude instrumental effects because all diagnostic data indicates stable operation for these runs. Therefore, we suspect variations of the glacier itself, i.e. spontaneous relaxation of cracks, refreeze of melting water within cracks during night as well as changes of the geometry of the melted holes including the water-level and the acoustic coupling of the [5]ensor and emitters to the bulk ice.

25.  In order to account for such changes in the error budget, we calculate the standard deviation $std(S_i)$. If this error is in excess of the previously estimated error from the mean of the repeated measurements it is added to the total error by $\sigma_{S,i}^2 = \sup(0, std(S_i)^2 - \sigma_n^2)$. [6]

**Page: 16**

Number: 1         Author: anonymous  Subject: Inserted Text        Date: 16.12.2018 01:50:32

,

Number: 2         Author: anonymous  Subject: Inserted Text        Date: 16.12.2018 01:50:33

,

Number: 3         Author: anonymous  Subject: Inserted Text        Date: 16.12.2018 01:52:41

is

Number: 4         Author: anonymous  Subject: Cross-Out  Date: 16.12.2018 01:53:21

Number: 5         Author: anonymous  Subject: Highlight   Date: 16.12.2018 01:55:09

use receiver consistently throughout the text

Number: 6         Author: anonymous  Subject: Highlight   Date: 16.12.2018 01:56:18

using the term total here is confusing because I take equation to represent the TOTAL error, not this.

Author: wiebusch   Subject: Notiz      Date: 28.12.2018 09:51:01

We think "total" is correct here, because this is what enters equation 1. to make it clearer we have reformulated and added "in Eq.1"

[Figure]

[Figure]

[Figure]

**Figure 6.** Measured amplitude for repeated measurements within series 7, 19 kHz

[Figure]

**Figure 7.** Measured amplitude for repeated measurements within series 11, 27 kHz.

**2.5.2 Reproducibility of measurements for repeated series**

To assess the reproducibility of full measurement series, three pairs of measurement series were taken between the same holes:
9 and 10 (10 m, directly consecutive), 6 and 24 (60 m, 4 days apart) and 15 and 19 (90 m, 1 day apart). In between, the setups
had been removed from their holes and then reinstalled.

5    Figure 8 shows the amplitude plotted against the frequency for all six measurement series. Overall, all three pairs show
a reasonably good consistency of the amplitude and shape of the curve within the estimated uncertainties. However, also
significant differences can be seen, e.g. for measurement series 6 and 24.

[Figure]

[Figure]

[Figure]

**Figure 8.** Amplitudes of measurement series 9 and 10 ($10\,\mathrm{m}$), 6 and 24 ($60\,\mathrm{m}$) and 15 and 19 ($90\,\mathrm{m}$).

[Figure]

**Figure 9.** Histogram of the relative variations between repeated measurements of the same hole pairs for all frequencies

In order to account for the variations in reproducibility we have investigated all measured relative differences $s_{nm} = (S_n - S_m)/(\sqrt{2}\cdot\overline{S_{n,m}})$. We find no dependency on the frequency and use the standard deviation $std(s_{nm}) = 0.45$ of this distribution (see Fig. 9) to account for the systematic uncertainty of time variations at fixed locations on the glacier $\sigma_{S_{n,m}} = 0.45 \cdot S_i$.

**2.5.3 Systematic differences related to different pairs of holes**

5 Figure 10 shows as an example the measured amplitudes as a function of the hole distance for $16\,\mathrm{kHz}$ sine bursts. The semilogarithmic plot displays a roughly linear dependency of amplitude and distance as expected. However, variations in amplitude exceeding the uncertainties of the individual measurements are visible at distances $20\,\mathrm{m}$, $40\,\mathrm{m}$ and $60\,\mathrm{m}$, see Table 2 for details

[Figure]

[Figure]

[Figure]

**Figure 10.** Normalized amplitudes 16 kHz sine bursts. Variations in measured amplitudes for measurements of different hole pairs at $20\,\mathrm{m}$, $40\,\mathrm{m}$ and $60\,\mathrm{m}$ are indicated.

[Figure]

**Figure 11.** Histogram of the relative difference between measurements of hole pairs of the same distance for all frequencies [1]

on the measurement series. Note, that this figure also displays the variations of repeated measurements of the same hole-pairs [2] $0\,\mathrm{m}$, $60\,\mathrm{m}$ and $90\,\mathrm{m}$) that are discussed in the previous section.

In order to estimate the uncertainty due to the propagation of signals through different ice masses, we have again investigated all relative differences of measured amplitudes of different hole pairs $(S_n - S_m)/(\sqrt{2} \cdot \overline{S_{n,m}})$ and estimated the standard deviation $std(s_n, s_m) = 0.68$. As this variation includes also the variation due to the time dependency [3] observed when using same holes, we subtract this respective uncertainty as estimated above $\sigma^2_{S_n, S_m} = std(s_n, s_m)^2 - \sigma^2_{S_{n,m}} = (0.68^2 - 0.45^2) \cdot S_i^2 = 0.51^2 \cdot S_i^2$.

Number: 1     Author: anonymous Subject: Inserted Text     Date: 16.12.2018 02:06:36

.

Number: 2     Author: anonymous Subject: Highlight   Date: 16.12.2018 02:07:36

This is not correct.

Author: wiebusch   Subject: Notiz      Date: 28.12.2018 09:57:12
Why do you think that this is this not correct ? We do not understand this remark.

Number: 3     Author: anonymous Subject: Cross-Out  Date: 16.12.2018 02:08:18

[Figure]

[Figure]

[Figure]

**Figure 12.** Measured propagation delay for $5\,\mathrm{ms}$ chirp signals

**Table 3.** Measurement of the propagation speed of sound $v_{prop}$.

|  | $v_{prop}$ | $\chi^2/n_{dof}$ |
| --- | --- | --- |
|  | m/s |  |
| chirp ($3\,\mathrm{ms}$) | $3443.0 \pm 0.2$ | $5.31/10$ |
| chirp ($5\,\mathrm{ms}$) | $3443.2 \pm 0.2$ | $5.31/10$ |
| chirp ($10\,\mathrm{ms}$) | $3447.9 \pm 0.2$ | $0.70/10$ |
| barker | $3477.0 \pm 0.1$ | $116.9/10$ |

**2.6 Speed of Sound Measurement**

An important verification of the *in situ* performance of the setup is the measurement of the speed of sound. For this measurement, we use the transmitted chirp and barker signals and estimate the propagation delay by the maximum correlation of emitted and received signals.

5    The used signals of $3\,\mathrm{ms}$ to $10\,\mathrm{ms}$ are shorter than the typical propagation delay of the acoustic wave. To avoid any influence of the electromagnetic induced signals, only measurements of distances larger than $10.8\,\mathrm{m}$ ($3\,\mathrm{ms}$), $18.0\,\mathrm{m}$ ($5\,\mathrm{ms}$) and $36\,\mathrm{m}$ ($10\,\mathrm{ms}$) are used as the signal emission is terminated before the acoustic signal reaches the receiver. The time-window of the electro-magnetic interference is excluded from the analysis.

The propagation delay is calculated by correlating for each measurement the recorded emitter voltage with the received

10    signal with a variable time offset. The time offset of maximum correlation determines the signal propagation time. The median from all repetitions of the same measurement is taken as well as the difference of the $15.85\,\%$ and $84.15\,\%$ quantiles for an estimate of the error.

[Figure]

[Figure]

The result of the measured propagation delay is summarized in Table 3 and shown in Fig. 12 for the example of $5\,\mathrm{ms}$ chirps. We observe a good linear behavior of the propagation delay with distance. From the chirp signals, a combined speed of sound of $(3444.7 \pm 1.6)\,\mathrm{m/s}$ is observed. The dominant systematic uncertainty on the absolute value of the speed of sound is related to the determination of the hole locations. The location of each hole has been measured with a GPS probe that showed a drift

5 of about $80\,\mathrm{cm}$ during the procedure. This drift corresponds to an uncertainty of about $30\,\mathrm{m/s}$.

The results for different chirps signals are, however, fully correlated with respect to this uncertainties and can be directly compared. The results of the $3\,\mathrm{ms}$ and $5\,\mathrm{ms}$ chirps are consistent with each other within their estimated fit-errors. The speed of sound derived from the $10\,\mathrm{ms}$ chirps deviates by about $5\,\mathrm{m/s}$ from those, and is thus not consistent within the errors that have been estimated from the fit. The barker signals show substantially stronger fluctuations in the propagation time which is

10 also reflected by a large $\chi^2$ value. The observed speed of sound deviates by $30\,\mathrm{m/s}$ from the results of the chirps. The barker signals are thus not taken into account in the further analysis.

We conclude, that the measured propagation delay sufficiently verifies the stability of the measurement setup. However, it also indicate not fully understood systematic uncertainties related to barker signals.

Our measured value of the speed of sound is smaller than $3880\,\mathrm{m/s}$ as measured for deep antarctic ice but larger than the

15 observations for firn ice (Abbasi et al., 2010). It is only slightly smaller than a previous measurement near the surface of alpine glaciers and antarctic glaciers with about $3660\,\mathrm{m/s}$ to $3700\,\mathrm{m/s}$ and $3500\,\mathrm{m/s}$ respectively (Helbing et al., 2016). However, there it was also observed that the propagation delay strongly depends on the direction and depth in the ice with variations up to $\pm 10\,\%$. This indicates a strong dependency on the structure of the ice and the morphology of the glacier. When taking into account these systematic uncertainties, we consider our observed value as a reasonably good confirmation of our measurement

20 procedures.

**2.7 Attenuation using Chirp signals**

The measured chirp signals can also be used to measure the attenuation of sound. For this, we have adopted a procedure that is mostly identical to the above described procedure in terms of estimation of uncertainties. Unlike the above procedure, the total received chirp signal as well as a noise window are Fourier transformed and the amplitude at the respective frequency is used

25 after noise subtraction. The Fourier transformation is recalculated for each frequency with a window length adjusted to this frequency in order to minimize spectral leakage. In comparison to the sine-burst measurement we do not measure a frequency clean signal and e.g. transient ringing of the receiver cannot be fully excluded from the measurement as easy. Furthermore, an uncertainty in the frequency dependency of the speed of sound and surface reflections may result in an uncertainty due to the dispersion of received signal. As the analysis of this data is thus less robust against these uncontrolled uncertainties we use this

30 independent data-set for a second measurement confirming our main result that is based on the sine-bursts.

As detailed for the measurement for the speed of sound, electromagnetic interference is no problem in case of chirps. Since the emission is terminated quickly, an overlap of the interference and the received acoustic signal happens only for short distances below $10.8\,\mathrm{m}$ ($3\,\mathrm{ms}$), $18.0\,\mathrm{m}$ ($5\,\mathrm{ms}$) and $36\,\mathrm{m}$ ($10\,\mathrm{ms}$) with an speed of sound of $3600\,\mathrm{m/s}$. As for the sine bursts, for all measurements up to distances of $20\,\mathrm{m}$ the electromagnetic interference is negligible due to the combination of high received

**Page: 21**

| | | |
|---|---|---|
| Number: 1 | Author: anonymous Subject: Inserted Text | Date: 16.12.2018 02:12:04 |

y

| | | |
|---|---|---|
| Number: 2 | Author: anonymous Subject: Inserted Text | Date: 16.12.2018 02:13:06 |

s

| | | |
|---|---|---|
| Number: 3 | Author: anonymous Subject: Inserted Text | Date: 16.12.2018 02:13:21 |

A

| | | |
|---|---|---|
| Number: 4 | Author: anonymous Subject: Inserted Text | Date: 16.12.2018 02:13:55 |

,

| | | |
|---|---|---|
| Number: 5 | Author: anonymous Subject: Inserted Text | Date: 16.12.2018 02:13:48 |

,

| | | |
|---|---|---|
| Number: 6 | Author: anonymous Subject: Inserted Text | Date: 16.12.2018 02:13:37 |

A

[revised manuscript text omitted]

**Page: 25**

Number: 1       Author: anonymous Subject: Cross-Out   Date: 16.12.2018 02:41:06

Number: 2       Author: anonymous Subject: Cross-Out   Date: 28.12.2018 10:15:57
brackets are correct as produced by siunix

Number: 3       Author: anonymous Subject: Cross-Out   Date: 16.12.2018 02:40:56

Number: 4       Author: anonymous Subject: Cross-Out   Date: 16.12.2018 02:41:23

Number: 5       Author: anonymous Subject: Cross-Out   Date: 16.12.2018 02:41:44

Number: 6       Author: anonymous Subject: Cross-Out   Date: 16.12.2018 02:41:44

Number: 7       Author: anonymous Subject: Cross-Out   Date: 16.12.2018 02:41:53

Number: 8       Author: anonymous Subject: Inserted Text       Date: 16.12.2018 02:42:00
, above which

Number: 9       Author: anonymous Subject: Inserted Text       Date: 16.12.2018 02:41:34
those

Number: 10       Author: anonymous Subject: Inserted Text       Date: 16.12.2018 02:42:10
,

Number: 11       Author: anonymous Subject: Cross-Out   Date: 16.12.2018 02:42:27

Number: 12       Author: anonymous Subject: Inserted Text       Date: 16.12.2018 02:42:22
,

Number: 13       Author: anonymous Subject: Inserted Text       Date: 16.12.2018 02:42:31
e

Number: 14       Author: anonymous Subject: Inserted Text       Date: 16.12.2018 02:42:53
do

Number: 15       Author: anonymous Subject: Inserted Text       Date: 16.12.2018 02:43:06
cause

Number: 16       Author: anonymous Subject: Inserted Text       Date: 16.12.2018 02:42:49
,

Number: 17       Author: anonymous Subject: Inserted Text       Date: 16.12.2018 02:43:27
cause

Number: 18       Author: anonymous Subject: Cross-Out   Date: 16.12.2018 02:44:30

Number: 19       Author: anonymous Subject: Inserted Text       Date: 16.12.2018 02:44:29
,

Number: 20       Author: anonymous Subject: Inserted Text       Date: 16.12.2018 02:44:35
e

Number: 21       Author: anonymous Subject: Inserted Text       Date: 16.12.2018 02:45:02
A

[Figure]

[Figure]

**Figure 15.** Comparison of our measurement to the results from Langleben (1969) for sea ice and Helbing et al. (2016). Shown are our results based on sine-bursts (dark grey band) as well as chirps (light red band) and the other reported results as data points

resulted in similar values. Further follow-up measurements on different glaciers would be required to confirm whether the effective attenuation of  a general property and whether it is related to the specific properties of ice.

In view of in-ice navigation of melting probes as described in Kowalski et al. (2016), our results confirm the possibility of the transmission of acoustic signals over tens of meters and thus allowing the determination of the position of a melting probe by the trilateration of acoustic signals. From our observations  frequencies below $20\,\mathrm{kHz}$ or even below $5\,\mathrm{kHz}$ are preferable for this application.

For the application of sub-glacial exploration, e.g. of deep sub-glacial lakes in Antarctica or a space mission to the moon Enceladus, the here observed attenuation would not allow for a navigation volume with sides much larger than typically $100\,\mathrm{m}$. However, the ice quality in other environments can be much improved.  Abbasi et al. (2011) observe an attenuation about $300\,\mathrm{m}$ in deep Antarctic ice. This would allow for a much larger propagation distance of sound and consequently a much larger navigation volume that scales with the cube of the maximum propagation distance. The feasibility of acoustic trilateration for the navigation in the ice shield of Enceladus remains an open question that depends strongly on the modeling of the local glacial environment. An ice-structure deviating from that of alpine glaciers could strongly enhance the performance of such a navigation system.

 presented measurement of the acoustic attenuation length is robust in terms of systematic uncertainties. The obtained values are encouraging for the development and the use of sonographic technologies for the exploration of natural glaciers, even in the presence of cracks and crevasses. An improved theoretical understanding of the effective damping of sound during propagation in such natural glaciers would allow determining whether the measured attenuation and its frequency dependency can be beneficial in characterizing basic properties of the glacier  ice.

| | Number: 1 | Author: anonymous | Subject: Inserted Text | Date: 16.12.2018 02:45:34 |

elastic energy

| | Number: 2 | Author: anonymous | Subject: Cross-Out | Date: 16.12.2018 02:46:22 |

| | Number: 3 | Author: anonymous | Subject: Inserted Text | Date: 16.12.2018 02:46:20 |

,

| | Number: 4 | Author: anonymous | Subject: Inserted Text | Date: 16.12.2018 02:47:24 |

length

| | Number: 5 | Author: anonymous | Subject: Inserted Text | Date: 16.12.2018 02:47:10 |

For example,

Author: wiebusch    Subject: Notiz    Date: 28.12.2018 10:27:58
removed e.g.

| | Number: 6 | Author: anonymous | Subject: Highlight | Date: 16.12.2018 02:47:37 |

at what frequency?

Author: wiebusch    Subject: Notiz    Date: 28.12.2018 10:30:49
added 10-30 kHz

| | Number: 7 | Author: anonymous | Subject: Inserted Text | Date: 16.12.2018 02:48:30 |

The

| | Number: 8 | Author: anonymous | Subject: Cross-Out | Date: 16.12.2018 02:49:32 |

| | Number: 9 | Author: anonymous | Subject: Inserted Text | Date: 16.12.2018 02:49:22 |

e

[revised manuscript text omitted]

**Resonse to Reviewers**

The authors

January 31, 2019

Dear Reviewers, dear Editor

we are very thankful for your thoughtful review and have tried to address all your comment in a satisfactorily manner. In this resubmission, you'll find the following documentation:

- The new, revised paper

- The same paper but with printed differences to the previous version

- Responses to your main comments

- As we have received many of your comments as acrobat inline, we have added a printout of the previous version with your responses and our comments on it.

We thank you for your valuable suggestions that have in our view substantially improved the paper We hope that this format is appropriate and look forward to your feedback.

Best regards

Christopher Wiebusch for the authors

**1 Response to reviewer #1, Henning Löwe**

1. *(p.1 l.6): here presented results → results presented here*
   ⇒ fixed, removed here
2. *& (p.2 l.9): polycrystaline*
   ⇒ fixed.
3. *(p4. l.21): maybe I missed it but when was the field campaign carried out?*
   ⇒ good catch. added "August 2017"
4. *(p.10 l.19): N → N*
   ⇒ fixed
5. *(p.11 l.27): what does sup stand for?*
   ⇒ Wiki: The supremum (abbreviated sup; plural suprema) of a subset S of a partially ordered set T is the least element in T that is greater than or equal to all elements of S, if such an element exists.Consequently, the supremum is also referred to as the least upper bound (or LUB).
   ...so it is the value is 0 unless the second argument is > 0, then it is the second.
6. *(Fig 6/7): should be combined to a single figure*
   ⇒ This is a difficult request, because this is distinct data that cannot be easily compared. The measurement time and span and measured amplitudes differ. We could put the figures side-by-side. We would need a specific suggestion how to combine - Otherwise we would prefer to keep it as it is.
7. *(p.15 l.4): a reference should be given for the used method*
   ⇒ fixed
8. *(p.20 l.11): the wave lengths (9 − 60 cm) as estimated from frequencies and measured speed of sound should be stated somewhere explicitly (not necessarily here, but the occurrence of wave length" reminds me of that) I think its helpful for the discussion later.*
   ⇒ Agree. We added right at the beginning that 1-100kHz corresponds to 350-3.5 cm
9. *(p.20 l.22): The statement about the comparison to Westphal in the frequency dependence is not clear. From which part of Fig 15 does this follow?*
   ⇒ Westphal is not included in the figure. The text is now modified to make this clear
10. *(p.20 l.24): I cannot follow why the present data is not consistent with Rayleigh scattering. Here it seems necessary to recall the prediction of Rayleigh scattering on the frequency dependence and maybe include an inset in Fig 15 to show how this compares to the collected data. In addition, the discussion and comparison to other work should be a bit more comprehensive in view of the similarities in view of of temperature, depth, ice porosity, etc. Given the range of wave lengths, the origin of attenuation by dissipative or scattering mechanisms may be quite different.*
    ⇒ For rayleigh scattering we would expect an attenuation length dependence

with the fourth power on the frequency. Our result is more in agreement with a dissipative loss, i.e. internal friction as suggested in the literature to be the dominating effect in warm ice.
We have improved the text by making it more comprehensive and clarer

11. *(p.20 l.29): Again, the conclusion about the frequency dependence is appears to be an overstatement if numbers (or*
*figures) are not shown.*
   ⇒ we have improved the text to be clearer here

12. *(p.20 l.32): accounts → account*
   ⇒ fixed

13. *(p.21 l.2): Isn't it possible to discuss/include at least the prediction of the attenuation coeffcient/length (maybe derived from the quality factor" as often used in the geo context) for homogeneous, polycrystalline ice in Fig 15?*
   ⇒ The predicted attenuation is at least wrong by a factor 10 as detailed in the introduction.

14. *(p.21 l.17): Acoustic scattering in heterogeneous materials is reasonably well understood, but it needs additional measurements to characterize the heterogeneities and the state of the material to infer potential origins.*
   ⇒ Yes, we agree. But it is not necessarily scattering. Changed to: An improved understanding of the effective damping of sound in natural glaciers is required before the attenuation and its frequency dependence can be beneficial in characterizing basic properties of the glacier ice. This will require to combine attenuation measurements with measurements of glacial parameters that characterize the heterogeneity and also to study temperature-dependent effects.

**2   Response to anonymous reviewer #2**

1. *However, I find the conclusions rather lacking, especially the comment regarding the attenuation mechanism as it relates to Rayleigh scattering. I think the authors would do well to reconsider this conclusion and really make an effort to discuss their reasoning and evidence for this conclusion...*

   ⇒ We agree and have substantially reworked the discussion.

2. *Pg 3 L14 (point 6): The water is necessary to propagate the compression wave. It is also there to keep the hole open I would assume and occurs no matter what because of the drilling method. I do not see the need for this statement. Why not just say water is present in the hole outside of this enumerated list? For instance on page 5 line 5 can be used for this.*
   ⇒ Added text: The water interface is advantageous compared to dry holes because it improves the coupling of the transducers to the ice.

3. *It would be great to have a map inset to see what in Italy this is located*
   ⇒ You can find it on google: `https://www.google.com/maps/place/Rifugio+Casati+al+Cevedale+mt+3269/@46.4703661,10.5718486,13z/data=!4m5!3m4!1s0x0:0xc0afb8a88f5d1295!8m2!3d46.463158!4d10.602489`
   We prefer not to change the figure to an even smaller scale., and the geographic locations are well defined.

4. *What was the surface-air temperature during these experiments?*
   ⇒ Outside air was up to +10 C during day but below 0 C during night.

5. *Paragraph structure (for example the first sentence in Section 2.4): A single sentence is not a paragraph. Please revise these sentences throughout the manuscript*
   ⇒ OK, we did throughout the document.

6. *Why is the electronic noise so strong? Did you use shielded cables? Was the excess cable wrapped in loops?*
   ⇒ Sure, cables are shielded and not looped, except for a few simple connectors. However, we generate 500V pulses for the largest probed distances inside the same DAQ box to which the signal comes back. We think, that the observed cross-talk on the few 10 mV level is actually really good.

7. *Pg. 9 L5: Is the crosstalk in the source signal as well? If it is, then how can you remove that cross talk from the amplitudes before you normalize?*
   ⇒ The cross talk is generated by the source signal. We do measure the amplitude of this signal and normalize the received acoustic signal to the emitted amplitude. This normalization is not affected by cross talk.

8. *Pg 9 L26: What does the following sentence actually mean? It does not make sense to me. "The noise estimate in the noise window matches the noise-level for the signal window reasonably well."*
   ⇒ Changed to: "The noise-level estimated from the noise window matches the apparent noise-level from the signal window reasonably well."

9. *Throughout document: Please use emitter and do not switch between emitter and*

*"sender." This is confusing. You do the same thing with sensor and receiver. Please stick to receiver.*

⇒ Fixed, thanks.

10. *Pg 10 last line: Where is the normalization by N to make this equation represent a mean? Also, the $\sigma_i^2$ terms cancel, so how is this an error weighted mean?*

⇒ These are standard text book formulas. The sigmas are inside the sum and do not cancel. If all sigma are the same, you get an division by N as expected for constant weights.

11. *Pg 16 L18: Is this variation due to fabric-induced anisotropy? If so, can you please discuss. The term "glacier geomorphology" is not very intuitive as it pertains to sound speed. I do not think readers will understand how geomorphology can cause velocity variations. I am not sure that I understand what you mean here.*

⇒ It is hard to interpret the origin of the effect that is reported in the cited reference. No definite statement can be done from our side. Sentence reads: "This indicates a strong dependency on the structure of the ice and the morphology of the glacier." and this seems to reflect the situation well.

12. *You discuss the influence of temperature changes on your measurements, but you do not cite recent and relevant work that studied attenuation as a function of temperature: "Monitoring the temperature-dependent elastic and anelastic properties in isotropic polycrystalline ice using resonant ultrasound spectroscopy", `https://www.thecryosphere.net/10/2821/2016/tc-10-2821-2016.html`*

⇒ Thank you for pointing us to that interesting reference. We have integrated it into the discussion.

13. *Your final comment on Rayleigh scattering in the conclusion section seems unfounded.*

⇒ That part is substantially reworked

14. *You reference the Westphal 1965 paper in your introduction, do some experiments, and then say, "look, we found it is not Rayleigh scattering". This is not rigorous, nor is it convincing. You pose no other mechanism and it seems like you would do the community a favor by providing a discussion as to why you think Rayleigh scattering is not the mechanism. Even explaining to the reader what Rayleigh scatter is would be a useful first step. Are you making this claim simply because your data do not follow an attenuation of frequency to the 4th power?*

⇒ Yes, that would be expected (see price et al.) . Maybe not a strict power of 4, to account for additional effects but a strong frequency dependence as is the claim in Westphal 1965 is not observed here. We have reworked the discussion.

15. *Please also note the supplement to this comment:*

⇒ Thank you for the very detailed review. All comments have been addressed. And with very few exceptions that are noted below we have followed your advice. For the full set comments please check the pdf with comments printed as well as the revised document with the highlighted changes.

16. *P8, L6: Change "After a delay of about 10 ms the acoustic signal sets in and is interfering with the cross-talk signal. → arrives (40 m from emitter) with an*

*amplitude above the electronic noise.*

⇒ This change would change the intention the sentence. Important is the interference of the signal with a fixed phase w.r.t. the electronic cross talk of the same frequency. Added "electromagnetic"

17. *P11, L26: using the term total here is confusing because I take equation to represent the TOTAL error, not this.*

⇒ We think "total" is correct here, because this is what enters equation 1. to make it clearer we have reformulated and added "in Eq.1"

18. *P14, L2: This is not correct.*

⇒ Why do you think that this is this not correct? We do not understand this remark.